# Research Progress on Elements of Wild Edible Mushrooms

**DOI:** 10.3390/jof8090964

**Published:** 2022-09-15

**Authors:** Shuai Liu, Honggao Liu, Jieqing Li, Yuanzhong Wang

**Affiliations:** 1College of Resources and Environmental, Yunnan Agricultural University, Kunming 650201, China; 2Institute of Medicinal Plants, Yunnan Academy of Agricultural Sciences, Kunming 650223, China; 3School of Agronomy and Life Sciences, Zhaotong University, Zhaotong 657000, China

**Keywords:** mushroom, trace elements, heavy metals, enrichment pattern, influencing factors

## Abstract

Wild edible mushrooms are distributed all over the world and are delicious seasonal foods, rich in polysaccharides, amino acids, vitamins, and other components. At the same time, they contain many essential trace elements and are highly enriched in heavy metals (compared to green plants and cultivated edible mushrooms). Consumers may be exposed to health risks due to excessive heavy metals in the process of consumption. This is also one of the important factors affecting the import and export of edible mushrooms, which is of great concern to consumers and entry and exit inspection and quarantine departments. In this paper, the contents of four essential trace elements of iron, manganese, zinc, and copper and four harmful heavy metals of cadmium, lead, mercury, and arsenic in nearly 400 species of wild edible mushrooms from 10 countries are reviewed. It was found that the factors affecting the elemental content of edible mushrooms are mainly divided into internal and external factors. Internal is mainly the difference in species element-enrichment ability, and external is mainly environmental pollution and geochemical factors. The aim is to provide a reference for the risk assessment of edible mushrooms and their elemental distribution characteristics.

## 1. Introduction

Large fungi capable of forming hard fungal tissue or large fleshy masses are called mushrooms and are widely distributed throughout the world [1,2]. Among the 16,000 recognized edible mushrooms, about 7000 can be eaten by humans, about 3000 are eaten by humans, and about 700 are considered to have certain medicinal value [3,4]. China is one of the world’s four oldest civilizations, and it is also the earliest country to recognize and take advantage of edible mushrooms. Its history can be traced back to the Yangshao culture period from 4000 BC to 3000 BC [5]. Wild edible mushrooms are also regarded as a delicacy in many countries in Eastern and Central Europe. Edible mushroom picking in the Czech Republic has become a “hobby of the whole people,” and statistics show that the average amount of edible mushrooms collected per household reaches 5.6 kg per year [6,7]. Wild edible mushrooms are popular products on the market and are extremely loved by consumers, not only because of their delicious taste but also because of their nutritional value [8]. Many vegans all over the world cannot obtain animal protein to supplement their nutritional needs, so edible mushrooms have become the best food resource for them to obtain protein. This is because of the high branched-chain amino acids obtained in mushrooms, which are generally found only in animal proteins. [9,10]. In some poorer developing countries, such as Nigeria, it is very difficult for most low-income families to obtain protein through beef and fish, so edible mushrooms have become a source of supplemental protein and an economic source for low-income families [11]. Edible mushrooms not only have nutritional value but also have certain medicinal value. Eating them every day can improve the immune system of the human body to achieve the effect of resisting diseases [12]. Mushrooms have been shown to have medicinal properties, and studies have shown that the beta-glucan in mushrooms may have a cholesterol-inhibiting effect and hypocholesterolemic activity [13,14]. Taking China as an example, Yunnan Province is rich in wild edible mushrooms and is the main source of wild edible mushrooms in China. The edible mushroom market is huge and is a major economic source for the local people. According to statistics, the total output value of edible mushrooms in Yunnan reached 660,000 tons in 2019, with wild edible mushroom production reaching 200,000 tons and artificially cultivated edible mushroom production reaching 460,000 tons, making a profit of RMB24.3 billion, RMB16.7 billion, and RMB7.6 billion, respectively [15]. The main representative product of the wild edible mushroom market in Yunnan is the Boletaceae. According to the China Edible Mushroom Association, in 2020, the production of Boletaceae in Yunnan Province was 89,363.2 tons, with a total output value of RMB281,261,82,000. (For more information, check the website http://bigdata.cefa.org.cn/output.html, accessed on 8 September 2022). Figure 1 shows the real situation of the edible mushroom market of Yunnan taken during a field survey. It reflects the important position of wild edible mushrooms in the market economy. Figure 2 shows the habitat map of wild edible mushrooms in Yunnan.

Trace elements are needed by the human body and play an extremely important role. For trace elements, too much or too little absorption will have different degrees of impact on the human body [16,17,18]. Heavy elements are always harmful to human health [19,20,21]. Mushrooms contain many essential trace elements and are highly enriched in heavy metals (compared to green plants and artificially cultivated edible mushrooms). There are many different species of edible mushrooms and significant differences in the levels of essential trace elements and heavy metals contained in the fruit bodies [22,23,24,25,26]. Soil background values and geochemistry are the main factors affecting the elemental content of fruit bodies, especially wild edible mushrooms growing near highways, industrial areas, chemical plants, smelters, etc., which often have excessive heavy metals [24,26,27,28]. In addition, different processing methods can also have some influence on the elemental content of fruit bodies. In this paper, the contents of four essential trace elements of iron, manganese, zinc, and copper and four harmful heavy metals of cadmium, lead, mercury, and arsenic in nearly 400 species of wild edible mushrooms from 10 countries are reviewed for the first time. The countries involved in the literature are shown in Figure 3. The purpose of this paper is to provide a reference for the risk assessment of edible mushrooms and their elemental distribution characteristics.

## 2. Content and Physiological Functions of Four Essential Trace Elements of Wild Edible Mushrooms

The amount of trace elements absorbed by the human body is limited, and too much or too little absorption will have different degrees of impact on the human body. When the amount of trace elements required by the human body exceeds the amount required by physiological functions, toxic effects will occur. At the same time, when the amount of trace elements absorbed is lower than the supply required by the human body, there will be different degrees of harm [29]. Therefore, when taking essential trace elements from edible mushrooms, uncontrolled intake will eventually lead to damage to body function. Wild-grown edible mushrooms are rich in protein and vitamins, and can supplement essential trace elements for the human body, which makes edible mushrooms become hot-selling products on the market and popular with consumers. Their fruit bodies contain not only a large number of essential trace elements but also harmful elements, such as lead, cadmium, mercury, and arsenic [30,31,32,33]. When people eat edible mushrooms, it is necessary to pay attention to the intake and not ignore the harm caused by improper intake of elements. When the dietary reference intake provided by the FAO/WTO is taken as the food guide, the intake of human nutritional elements is guaranteed to be within the safe range. The second section provides an overview of the elemental contents of Fe, Cu, Zn and Mn in different species of edible mushrooms from different regions (note: the elemental content values mentioned in the article are all dry-matter content). Examples of the physiological effects of the elements on the human body are also given, as shown in Table 1. Food is so closely related to the human body that understanding the role of its own elements can only be truly valued in a meaningful way.

### 2.1. Physiological Role of Iron and Content of Iron in Edible Mushrooms

Physiological role: Iron is an indispensable component for the growth and development of plants, animals, and humans. It is a fact that plant fruit bodies contain iron that enables humans to absorb iron from food, which is also the main channel for humans to obtain iron. Because trace elements cannot be produced in the human body through physiological effects, they can only be obtained from food, air, and exogenous substances. It is for this reason that the iron content in the human body is lower or higher than the normal value required by the human body [42]. Seventy percent of the body’s iron is found in hemoglobin, other iron compounds make up only a small fraction, myoglobin makes up four percent, and most of the extra iron is found in several enzymes. This shows that iron is an important component of hemoglobin, myoglobin, and oxidase [34]. When the body’s iron levels are lower than normal, this can cause diseases, mainly iron-deficiency anemia. Iron deficiency leads to a decrease in the number of red blood cells, resulting in a decline in oxygen transport function, and ultimately resulting in hypoxia of human organs [18]. Iron exists in the human body in many forms and has a wide range of functions. Iron-deficiency anemia is caused by the inhibition of deoxyribonucleic acid synthesis in the liver when the iron is deficient, thereby hindering the development of the liver. This affects mitochondria and microsomes in liver cells and other cells, resulting in reduced protein formation and energy use, ultimately leading to anemia, along with symptoms such as incomplete height and weight development. Iron poisoning occurs when the iron element in the human body exceeds the normal level. The cause may be related to food. Iron contained in ingested food can be absorbed in large quantities, but it may come from changes in the human body [17]. These are only some aspects of the role of iron in the human body, but this is enough to attract people’s attention. While pursuing physical health, one should not consume too much or too little iron. A relatively balanced state is the safest and healthiest.

A total of 10 species (Table 2) of wild edible mushrooms were selected by Isildak et al. [43] in Tokat, in the middle Black Sea region of Turkey. The habitat of each mushroom was different and the iron content was determined to be 174 ± 15.9 mg/kg to 335 ± 26.5 mg/kg. The difference of almost 200 units between the lowest and highest values of Fe content may be due to the different species studied. In addition, some differences in habitat could be the reason for the difference. Sarikurkcu et al. [44] studied wild edible mushrooms from hazelnut orchards in Arsin, Trabzon, Turkey. The study area had low involvement of human activities and was not polluted by industrialization. A total of 26 species (Table 2) were studied and the iron content ranged from a minimum of 79.00 ± 1.00 mg/kg to a maximum of 7769.00 ± 87.00 mg/kg. The difference in iron content was more significant than in the study by Isildak et al. [43]. Such a large variation in iron content in the absence of almost any human involvement and pollution is most likely the result of species specificity. The study area selected by Ouzouni et al. [45] was rangeland away from potential sources of pollution and uncontaminated forest in Greece. Ten species (Table 2) of wild edible mushrooms were selected, with greater or lesser variation in habitat. The iron content ranged between 38.90 ± 1.97 mg/kg and 499.0 ± 7.82 mg/kg. The iron content of *Amanita caesarea* was determined to be 365.90 ± 4.46 mg/kg. In addition, the iron contents of *A. caesarea*, *A. ceciliae*, and *A. vaginata* determined by Sarikurkcu et al. [44] were 114.00 ± 5.00 mg/kg, 2659.00 ± 26.00 mg/kg, and 1563.00 ± 9.00 mg/kg, respectively. A comparison of the content values in the two papers showed that even if the same genus was grown in an unpolluted environment, differences in iron content could still occur [44,45]. The difference is due to the different growing environment, which may be a reason to influence the Fe content. In addition, although belonging to the same genus, the different species may have some influences. The growth of wild edible mushrooms in large urban areas of Athens was the subject of a study by Kokkoris et al. [46]. The significance of this is that the majority of the population of Athens lives in the metropolitan area, where human activities are extensively involved, and it is the most polluted area [47]. The selected sites were road edges, public squares, playgrounds, unbuilt spaces, and parks. The authors studied a total of six species (Table 2) of wild edible mushrooms with a minimum iron content of 275.46 ± 75.85 mg/kg and a maximum of 736.83 ± 212.74 mg/kg. Kokkoris et al. determined that the lowest values of iron content were higher than the lowest values measured by the previous researchers [43,44,45,46]. The difference in the study context is that the metropolitan area of Greece is polluted, while the other studied areas are not significantly polluted. This could be the reason that the minimum values of iron content were higher than the other minimum values. Most notably, both Kokkoris et al. and Isildak et al. [43] measured *A. bisporus* and found significant differences in iron content values. A comparative graph of iron content is shown in Appendix A.

Reference standard for daily intake of human iron: According to the European Union’s daily intake of trace elements for humans, the appropriate intake of iron is 11 mg/d for men 18–70 years old and 13 mg/d for women 18–70 years old [48]. The United States has established a daily human trace element intake of 8 mg/d for men 18–70 years old, 18 mg/d for women 18–50 years old, and 8 mg/d for women 50–70 years old. The maximum tolerable intake of iron for both men and women is 45 mg/d [49]. China has established a daily human trace element intake value of 12 mg/d for iron for men 18–50 years old and 20 mg/d for women 18–50 years old. The maximum tolerable intake of iron for both men and women is 42 mg/d [50].

### 2.2. Physiological Role of Copper and Content of Copper in Wild Edible Mushrooms

Physiological role: Compared to elements such as iron and zinc, no special attention has been paid to physical function problems caused by copper deficiency, resulting in relatively weak knowledge about copper [16]. As one of the essential trace elements in the human body, copper plays an important role in the physiological functions of the human body [18]. Therefore, it is of great significance for consumers to understand the importance of copper in the human body. Generally, the content of copper in normal adults is between 100 mg and 200 mg, mainly in the liver, blood, muscles and bones. The copper content in the blood accounts for 5–10% of the total copper content in the human body, the liver accounts for 20% of the total human body content, and the largest proportion in muscles and bones, reaching 50–70%, with little in the copper enzymes [16]. In the human body, the most common cause of copper deficiency is the imbalance between the needs of human physiological functions and the supply of dietary copper [51]. In the blood and nervous system, copper is extremely important as an essential trace element for the growth and formation of bones and myelination of the nervous system. Copper helps hemoglobin absorb iron, and the gastrointestinal tract absorbs iron and transfers iron from tissue to plasma. Therefore, insufficient copper in the human body can also lead to symptoms such as anemia [35]. Osteoporosis occurs in most patients with severe copper deficiency [36]. The daily copper intake of normal adults is 2–5 mg, of which about 40% is absorbed into the blood in the intestine and most is absorbed in the liver. When the human body excretes copper, more than half of it is excreted through bile, and a small part is excreted in the form of urine [52].

Mironczuk-Chodakowska et al. [53] studied wild edible mushrooms grown in the “green lung area” of Poland, with a total of 20 species (Table 3) selected for determination of elemental copper content. The green lung area has a small population, low urbanization, and no heavy industry. A lowest value of 7.3 ± 2.41 mg/kg and a highest value of 123.0 ± 34.9 mg/kg were obtained for the 20 species of mushrooms. Eighteen species can be a good source of copper supplementation for adults and do not pose a toxicological hazard to humans. Keles et al. [54] collected 20 species (Table 3) of wild edible mushrooms in Erzincan province, Turkey. The values of copper content were determined and found to be a minimum of 4.02 mg/kg and a maximum of 128.94 mg/kg. *Leccinum versipelle* and *Russula delica* had the highest copper content, with 102.4 mg/kg and 128.94 mg/kg, respectively. No other species of mushroom, except *Mitrophora semilibera*, contained more than 50 mg/kg of copper, which may be due to species differences. Sesli et al. [55] determined copper in 14 species (Table 3) of wild edible mushrooms growing in the Black Sea region of Turkey. Values were between 15.5 ± 1.3 mg/kg and 73.8 ± 5.6 mg/kg. Mushrooms containing elemental copper values in the range of 100 to 300 mg/kg dry matter are not considered to be a health risk [56]. This shows that these 14 species of wild edible mushrooms in the Black Sea region of Turkey are safe for consumption and pose no toxicological hazard to humans. Fu et al. [57] determined copper elements in 19 species (Table 3) of wild edible mushrooms growing in Yunnan Province, China. The copper content ranged from a minimum of 10.3 ± 0.2 mg/kg to a maximum of 55.1 ± 1.4 mg/kg, which also indicates that the 19 wild edible mushrooms studied do not pose a toxicological hazard to humans [56]. By comparing the 71 species of wild edible mushrooms from the above four countries, it was found that the differences in copper values were not very significant. The maximum values did not exceed 300 mg/kg and did not pose toxicological hazards to humans [53,54,55,56,57]. Nevşehir and Niğde are two small cities in the tourist area of Cappadocia, Turkey. A total of 16 species of wild edible mushrooms were collected for the determination of copper content in both locations by Narin et al. A total of 12 of these species were measured in Nevşehir, with copper content ranging between 6.7 ± 1.1 mg/kg and 250.0 ± 23.0 mg/kg. The copper content of 11 of the wild edible mushroom species was determined in Niğde, and the lowest value of 58.0 ± 4.0 mg/kg and the highest value of 1353.0 ± 32.0 mg/kg was obtained [58]. Twelve species of wild edible mushrooms grown in the Nevşehir region did not contain more than 300 mg/kg of copper, while 11 species from the Niğde region contained a maximum of 1353 mg/kg of copper, which could be harmful to humans [56]. A comparative graph of copper content is shown in Appendix A.

Reference standard for daily intake of human copper: The European Union recommends a daily intake of 1.6 mg/d and 1.3 mg/d for adult (18–70 years old) men and women, respectively [48]. The United States has established a daily intake of 0.9 mg/d for adult (18–70 years old) men and women, with a tolerable maximum intake of 10 mg/d [49]. The standard for China is daily intake of copper for adult (18–50 years old) men and women of 0.9 mg/d, with a tolerable maximum intake of 10 mg/d. The standard in China is 0.8 mg/d for both the appropriate daily intake and the tolerable maximum intake of copper for adults (18–50 years old) [50].

### 2.3. Physiological Role of Zinc and Content of Zinc in Wild Edible Mushrooms

Physiological role: Zinc is the most widely metabolized element in the human body, and is a component of deoxyribonucleic acid (DNA), ribonucleic acid (RNA), and more than 200 kinds of enzymes in the human body. If the zinc element in the human body is lower than the level required for normal physiological functions, the enzyme activity in the body will be reduced, which will lead to the failure of multiple systems in the human body to perform normal physiological functions [59]. Zinc mainly exists in the bone and skeletal muscle of the human body and is an important component of most zinc-dependent enzymes, especially in the synthesis and degradation of lipids, proteins, and nucleic acids [37]. Once the essential minerals and vitamins needed by the human body are lower than the amount required by the normal physiological function of the human body, we call it “hidden hunger.” The phenomenon is often imperceptible to the human body, resulting in an impact on physical health. Globally, more than 2 billion people are suffering from hidden hunger [60]. The severity of the health problems caused by zinc in developing and developed countries is completely different. The health problems caused by zinc in developing countries are much higher than in developed countries, mainly due to insufficient zinc from food [61]. There is no special mechanism for storing zinc in the human body. Zinc cannot be supplied at any time like human energy. The only way to supply it is to continuously ingest it from the outside world. Food is the main way for the human body to get zinc, so the zinc content in food has become the main factor that affects the absorption of zinc in the human body. If the zinc content in food is low, it is difficult to meet the needs of normal physiological functions of the human body [62]. Zinc itself is not highly toxic, but when a large amount of zinc, i.e., 1 g, is absorbed, it will also have certain effects on the human body, such as nausea, vomiting, diarrhea, fever, lethargy, and other acute symptoms. However, when a small amount of zinc exceeds the reasonable intake of the human body, the internal balance can be adjusted by increasing the excretion of endogenous feces and urine. Excessive intake of zinc for a long time will affect the absorption of other trace elements, especially copper [38]. When zinc is deficient, the growth, sexual maturity, and immune defense system will be negatively affected [37].

Zinc is widely present in living organisms and has some biological significance. Mushrooms are known as zinc accumulators and the sporophore [63]. Genccelep et al. [64] determined zinc in 30 species (Table 4) of wild edible mushrooms grown in Erzurum province, Turkey, with the lowest value of 26.7 mg/kg and the highest value of 185 mg/kg. Ouzouni et al. [65] studied eight species (Table 4) of wild edible mushrooms from forests in Epirus and West Macedonia, Greece. The measured zinc content ranged between 35.9 ± 1.8 mg/kg and 96.9 ± 4.5 mg/kg. Siric et al. [66] determined the zinc content of 10 species (Table 4) of wild edible mushrooms growing in the Medvednica Nature Park (located near the capital, Zagreb, the largest urban and industrial center of Croatia). The Zn content of the soil in the study area was 79.07 mg/kg, while the lowest Zn content of 41.99 mg/kg and the highest Zn content of 90.56 mg/kg were obtained. By comparing the minimum values of zinc in a total of 48 wild edible mushrooms from three countries (Turkey, Greece, and Croatia), it was found that Croatia had the highest minimum values of zinc in wild edible mushrooms. This may be due to the environmental contamination of the study area and the high values of soil zinc content. In addition, mushrooms are known as zinc accumulators and the sporophore [63,64,65,66]. Chowaniak et al. [67] determined the zinc content of *Lactarius salmonicolor* and measured zinc content of the soil in the environment in which it was grown. The results revealed that there was a significant correlation between the Zn contents in both. This suggests that mushrooms have the potential to serve as a monitoring tool for environmental contamination, reflecting a level of soil contamination laterally through the elemental content of mushrooms. Studies on mushrooms as biomonitors or good bioindicators of environmental pollution have confirmed that this is possible [68,69,70,71]. Keles et al. [72] collected 20 species (Table 4) of wild edible mushrooms from towns near Gumushane province, Turkey. Zinc content was determined to be between 22.99 mg/kg and 91.76 mg/kg. Compared to the first three ranges of zinc content of wild edible mushrooms, they do not differ much from each other in terms of the lowest values of zinc content. When comparing the highest values of content, it was found that the wild edible mushrooms determined by Genceelep et al. had the highest value of zinc content, differing from the other three categories by almost 90 units [64,65,66,72]. Mushrooms are inherently good accumulators of zinc and are able to accumulate a certain amount of zinc without significant contamination [63]. This suggests that, in addition to the properties of the mushrooms themselves, environmental factors may be an important factor influencing the accumulation of zinc content. A comparative graph of zinc content is shown in Appendix A.

Reference standard for daily intake of human zinc: According to the United States, the standard for appropriate daily intake of zinc for adults 18–70 years old is 11 mg/d for men and 1.8 mg/d for women. For adults, the tolerable maximum intake is 400 mg/d [49]. The recommended daily intake of zinc in China is 12.5 mg/d for men and 7.5 mg/d for women in adults 18–50 years old. The maximum tolerable intake for adults is 40 mg/d [50].

### 2.4. Physiological Role of Manganese and Content of Manganese in Wild Edible Mushrooms

Physiological role: Manganese is also one of the essential trace elements for the normal operation of the human physiological system. Its content in the human body ranges from 12 to 20 mg, mainly concentrated in the brain, kidney, pancreas, and liver. Manganese can increase the rate of intracellular fat oxidation, thereby improving fat metabolism in patients with atherosclerosis, reducing the accumulation of liver fat, and protecting the cardiovascular and cerebrovascular systems of the elderly [18,39]. The active part of many enzymes in the human body contains manganese, which plays a direct role in the body’s immunity and indirectly affects the body’s antiaging ability [40]. Too high or too low an intake of manganese will cause different degrees of negative effects on the human body. When the body is deficient, skeletal deformity is one of the manifestations, and when the human body accumulates excessive manganese, it will cause human dysfunction [41].

Isiloglu et al. [25] collected six species (Table 5) of wild edible mushrooms in Balikesir, in the northwestern part of Turkey. The mushrooms were taken from an area near a highway and a park area. The lowest value of manganese content in mushrooms near the highway was 14.5 mg/kg and the highest value was 63.6 mg/kg, while the manganese content of mushrooms in the park area ranged between 14.5 mg/kg and 52.6 mg/kg. Among the mushrooms harvested from the highway side, only *Lactarius sanguifluus* had more than 60 mg/kg of manganese, while the others were less than 31 mg/kg. The manganese content of *L. sanguifluus* in the mushrooms grown in the park was only 14.5 mg/kg, but *Clitocybe alexandri* and *C. flaccida* both contained about 52 mg/kg. This may be the difference caused by different growing environments. Gezer et al. [73] collected six species (Table 5) of wild edible mushrooms growing in a forest area near the city of Denizli, Turkey, and determined the manganese content of the mushroom fruit bodies and soil. Manganese content in the soil ranged from a minimum of 98.7 ± 11.3 mg/kg to a maximum of 212.4 ± 13.5 mg/kg, while there were no significant differences in manganese content among the six species of mushrooms. This shows that even high levels of manganese in the soil did not affect the content in mushrooms excessively. Ouzouni et al. [74] collected eight species (Table 5) of wild edible mushrooms from forests in West Macedonia, Greece. The measured manganese content ranged from 8.57 ± 0.44 –35.1 ± 1.6 mg/kg. Manganese elements were determined in seven species (Table 5) of edible mushrooms grown in the western region of Turkey by Tel-Cayan et al. [75]. The lowest value of manganese element was 4.92 ± 0.14 mg/kg and the highest value was 76.4 ± 2.84 mg/kg. Referring to previous studies, the elemental content of manganese is not considered high when compared to the elemental content of iron, copper, and zinc [43,44,53,54,65,66,74,75]. The lowest and highest values were not significantly different from each other. Even if the soil surface contained high values of manganese, the content of the fruit bodies was not affected significantly [25,73,74,75]. There are exceptions, however. Li et al. [76] selected 20 different sites in China to investigate the values of elemental manganese in *Tricholoma matsutake* and soils. The Mn content of the fruit bodies was much higher than that in the soil. A comparative graph of manganese content is shown in Appendix A.

Reference standard for daily intake of human zinc: The European Union recommends a daily intake of 3 mg/d of manganese for adults 18–70 years old [48]. In the United States, the recommended daily intake of manganese for adults 18–70 years old is 2.3 mg/d for men and 1.8 mg/d for women, and the tolerable maximum intake for adults is 11 mg/d [49]. In China, the recommended daily intake of manganese for adults 18–50 years old is 4.5 mg/kg, and the tolerable maximum intake is 11 mg/d [50].

## 3. Physiological Functions and Content Physiological Functions of Four Heavy Elements of Wild Edible Mushrooms

Heavy metal contamination has been a hot topic, and there are many researchers who have determined the significance of heavy metal levels in edible mushrooms. Once heavy metal levels in edible mushrooms are exceeded and enter the body through the food chain, the damage caused is irreversible [77,78,79]. There are several definitions of heavy metals, and it is generally accepted that metallic elements with a density of 4.5 or more and with an atomic number greater than 11 that pose a threat to the environment and humans are called heavy metals, which include two elements—arsenic and selenium. Heavy metals, such as the elements cadmium, lead, mercury, and arsenic, can pose a threat to human physiological functions and affect health when they are present in the body above acceptable levels [20,21]. The International Cancer Center and the US Environmental Protection Agency’s Integrated Risk Information System divide heavy metals into potentially toxic chemical carcinogens, such as lead, cadmium, mercury, arsenic, and biologically essential nonchemical carcinogens [80]. If humans are exposed to lead, cadmium, mercury, and arsenic for a long time, it will cause serious harm to the body. Human exposure is generally through inhalation of air pollutants, drinking contaminated water, and eating contaminated food [81,82]. These are also due to human production activities, resulting in the atmospheric environment, water environment, and soil environment pollution, indirectly resulting in plants absorbing more pollutants, and ultimately absorbed by the human body, resulting in a series of health and safety problems. The maximum intake values for heavy metals established by the WTO/FAO are 0.1 mg/kg for cadmium, 0.2 mg/kg for lead, 0.03 mg/kg for mercury, and 0.1 mg/kg for arsenic [83]. The third part of this review focuses on the levels of Cd, Pb, Hg, and As in different species of edible mushrooms. The effects caused by the four harmful elements on humans are listed in Table 6.

### 3.1. Physiological Role of Cadmium and Content of Cadmium in Wild Edible Mushrooms

Physiological role: Cadmium, as a heavy metal, has no known biological function in the human body, so it is regarded as a toxic heavy metal element that is unfavorable to the human body [87]. Among the metal elements, cadmium has its unique side. It has many toxic effects and a long biological half-life, which is about 20 to 30 years in the human body [95]. Once the cadmium element enters the human body, it will exist in the human body for a long time, mainly in the kidneys and liver. Only some cadmium will be slowly excreted in the urine and feces. Too much cadmium will overload the liver and kidneys, which will still cause harm to the human body [88]. Cadmium can also cause damage to male reproductive function and causes certain toxicity to the testes. Due to the development of human beings, the phenomenon of the soil environment being polluted is relatively serious. Cadmium pollution in soil is one such concern. Plants growing under cadmium-contaminated soil contain cadmium to varying degrees. When these plants are ingested by humans and animals, cadmium accumulates in organs such as the liver and testes [19,87]. There is also a certain difference in the absorption of cadmium between men and women. The absorption of cadmium in women is higher than that in men, which is the result of low serum ferritin value due to the reduction of iron reserves in women [96].

Melgar et al. [97] collected 13 species (Table 7) of wild edible mushrooms from the province of Lugo, northwestern Spain, to determine cadmium levels in the hymenophore and rest of the fruit body. These data were related to several factors: species and ecology (mycorrhiza and humus), morphological (hymenophore and the rest of the fruit body), and the influence of traffic pollution. The cadmium content of the hymenophore was higher than that of the rest of the fruit body in all cases. The highest value of cadmium was found in the hymenophore of *A. macrosporus*, with a mean value of 68.96 mg/kg. The second was *M. procera*, with 1.57 mg/kg. The highest values of specific proteins bound to elemental cadmium were found in *A. macrosporus*, so the authors inferred that this may not have much to do with traffic contamination, but rather due to the species. It is also suggested that *A. macrosporus* is not recommended for consumption. Orywal et al. [98] determined the elemental cadmium content of the two most commonly consumed mushrooms (Table 7) in Poland and the value of cadmium was higher in *B. edulis* (1.983 mg/kg) than in *Xerocomus badius* (1.154 mg/kg) (note: it is the same species as *Imleria badia* in Table 3, as it is a synonym of *Imleria badia*). This difference was statistically significant and in agreement with the data provided by other researchers. In another study, *B. edulis* contained 1.840 mg/kg of cadmium, more than twice as much as *X. badius* (0.746 mg/kg) [99]. The authors concluded that *B. edulis* may pose some risk to humans due to cadmium ingestion and that wild mushrooms should be monitored for harmful elements [98]. Ndimele et al. [100] determined cadmium levels in substrates and seven wild edible mushroom species (Table 7) at five sites (roadside, park, lawn, forest, and highway). The highest cadmium values were found in *V. speciosa* grown on the highway and the lowest values were found in *C. melliolens* grown on the lawn. The authors compared the measured values with other studies with a large difference in cadmium elements. It was concluded that the accumulation of heavy metals by mushrooms depends on the species as well as the heavy metal content of the substrate. Gao et al. [101] determined the values of cadmium in the caps and stipes of eight species (Table 7) of wild edible mushrooms in Yunnan Province, China. It was found that the caps were more likely to have a higher elemental content than the stipes. The authors posited that the reason might be that the cap was the first to be exposed to atmospheric heavy metal deposition or that the heavy metals accumulated in the cap through the bottom-up movement of the mycelium [102]. The results of cadmium element determination showed significant differences (*p* < 0.05) among different wild edible mushroom species and different parts. Among them, *C. ventricosum* had the highest health risk. When eating and collecting wild edible mushrooms, mushrooms grown in areas contaminated with heavy metals should be avoided. We should avoid eating and picking mushrooms that are grown in environmentally contaminated areas and have a high capacity for heavy metal enrichment due to the characteristics of the species. [97,98,100,101]. A comparative graph of cadmium content is shown in Appendix A.

### 3.2. Physiological Role of Lead and Content of Lead in Wild Edible Mushrooms

Physiological role: Lead is one of the heavy metals causing serious environmental pollution. It is hidden in the air, water, and food, posing a huge threat to human health [84]. The majority of lead is retained in the bones of adults, and the concentration of lead in bones increases with age. Children do not retain lead in their bones like adults [85]. Only a small part can be excreted by the human body, and the human body excretes lead in two ways—urine and feces. It is excreted in urine in a larger proportion [86]. Lead affects the normal transmission of neuronal signals because lead mimics calcium, or competes with calcium. After successfully competing with calcium, lead occupies the binding site of cerebellar phosphokinase, which in turn affects neuronal signal transduction. High levels of lead in the human body can affect the reproductive system of both men and women. For men, it will cause abnormal prostate function, greatly reduce sperm count and vitality, and reduce libido, while for women, lead poisoning will increase the miscarriage rate of pregnant women, resulting in infertility or premature birth in normal women [103]. In addition, some important iron-dependent metabolic processes are also affected by lead, such as the biosynthesis of heme, resulting in anemia and impaired cognitive development, with serious consequences for children [104]. Most of the sources of lead in humans and animals are ingested from food, so it is extremely important to understand the element content in food [20].

Zhu et al. [105] collected 14 species (Table 8) of wild edible mushrooms in Yunnan Province, China, and studied their heavy metal contents. The lowest elemental lead content was 0.67 ± 0.05 mg/kg and the highest was 12.9 ± 1.0 mg/kg. The authors found that only *P. eryngii* contained elemental lead above the legal limit through comparative studies. However, other mushrooms do not contain only one element in excess of the standard. Therefore, it is recommended that the heavy metal content of wild edible mushrooms be analyzed to assess the potential hazard to human health. The heavy metal content of 15 wild edible mushrooms (Table 8) growing in forests near historical silver-mining areas was determined by Svoboda et al. [106]. The minimum value of elemental lead was 3.15 mg/kg and the maximum value was 37.6 mg/kg. The authors analyzed the effect of the environment on enrichment of wild mushrooms with heavy metals. It was considered that because the life span of the fruit bodies was short, usually only 10–14 days, the effect of atmospheric deposition was excluded. Wild edible mushrooms grown at this study site have high levels of lead and should be restricted in terms of consumption. The authors speculate that other mushrooms grown in areas similar to this one may also have excessive levels of heavy metals. Komarek et al. [107] collected three wild edible mushroom species (Table 8) near a heavily polluted secondary lead smelter and measured the elemental content values. The soil in the study area was also measured and found to have high concentrations of elemental lead. Comparing the three mushrooms, *B.edulis* had the highest elemental lead content, followed by *X. badius* and *X. chrysenteron*. The authors concluded that the metal content related to species and age, increasing with the age of the fruit bodies. Ivanic et al. [108] studied the possible effects of urban pollution on the uptake of heavy metal elements by wild edible mushrooms. Fifteen and four, species (Table 8) of wild edible mushrooms were collected in urban and forested areas exposed to urban pollution, respectively. The lead content of wild edible mushrooms in urban areas ranged from 0.072 mg/kg to 1.11 mg/kg, and 0.18 mg/kg to 4.019 mg/kg in forest areas. The authors measured the highest levels of elemental lead even in forested areas far from major pollution sources, although they did not exceed the usual levels for mushrooms. This is mainly influenced by the geochemistry of the underlaying soil and by the nature of the mushroom itself. By comparing previous studies, the heavy metal content of wild edible mushrooms was affected by atmospheric deposition, geochemistry of soil, life cycle, species and industrial pollution, but mainly by species, geochemistry of soil, and environmental pollution [105,106,107,108]. In addition, when analyzing the effect of mushroom age on metal content, Svoboda et al. [106] concluded that mushrooms live only 10–14 days and that atmospheric deposition does not have a significant effect on metal content. In contrast, Komarek et al. [107] concluded in their study that the metal content increases with the age of the fruit bodies, which does not conflict with each other. Although both are considered in terms of life-cycle length, the former considers atmospheric deposition and the latter considers soil geochemistry. A comparative graph of lead content is shown in Appendix A.

### 3.3. Physiological Role of Mercury and Content of Mercury in Wild Edible Mushrooms

Physiological role: Mercury levels above 8 mg/L in the human blood can reduce sperm quality and fertility [89,90]. Alkylmercury is much more toxic than soluble inorganic mercury, causing damage to the human nervous system, proteins, and nucleic acids [91]. Mercury will accumulate in the central nervous system, digestive system, and kidneys. It is harmful to the human respiratory system, skin, blood, and eyes [109].

Demkova et al. [110] collected a total of 501 samples of three wild edible mushroom species (Table 9) from 60 regions in Slovakia. The Slovak limit for soil mercury was exceeded at 22 of the 60 sampled sites, mainly in former mining areas. There was a strong correlation between the evaluated parts for all samples, the strongest correlation being confirmed between the cap and the stipe of the mushroom. The results indicate that mushrooms accumulate risk elements from the growing environment and that all parts of the mushroom are affected. The authors found that *B. subtomentosus* was the best accumulator of mercury based on bioaccumulation factors. From the results, it was concluded that there is a great risk of consuming mushrooms grown in areas classified as environmentally polluted. Arvay et al. [111] collected five species (Table 9) of wild edible mushrooms from historical mining and processing areas around Slovakia. Determination of elemental mercury levels in soil substrate and fruit bodies revealed the highest bioconcentration factor for *M. procera*. Even though the level of elemental mercury in the soil substrate where it was located was not high, the elemental mercury content of the fruit bodies was still the highest. The highest value of elemental mercury was 1.98 ± 68.2 mg/kg and the lowest was 0.15 ± 41.0 mg/kg for the five species of wild edible mushrooms. The authors concluded that species like *M. procera*, which have high bioaccumulation capacity, may pose a threat to humans if consumed. Vetter et al. [112] determined the elemental mercury content of a total of 112 fruit bodies of 36 species (Table 9) of wild edible mushrooms grown in Hungary. Among them, *L. inversa* growing in *halmi* wood and *L. luscina* in *kamara* wood had the highest elemental mercury content of 7.44 mg/kg and 9.88 mg/kg, respectively. The reason is that both of them are close to the center of Budapest, which has a high level of environmental pollution. By comparing the values of the elemental mercury content contained in the different species, significant differences were found. The study area selected by Arvay et al. [113] is a volcanogenic area where geochemical anomalies occur frequently and penetrate the surface. The substrate consists of high levels of elements such as mercury. The mushrooms grown in such an environment are severely overloaded with elemental mercury. In other words, mushrooms represent an integral part of the environment. Severe environmental contamination can be demonstrated in the fruit bodies of mushrooms. The authors collected six species (Table 9) of wild edible mushrooms for heavy metal content monitoring, which was also an investigation of the level of environmental pollution. The results showed that the lowest value of elemental mercury was 13 mg/kg and the highest 52 mg/kg. The pollution of the environment by elemental mercury does not occur only in some specific circumstances. Excess levels of elemental mercury in the soil can still be detected in forests far from industrial areas, volcanogenic zones, mining areas, etc. [110,112]. This phenomenon is most likely due to the airborne transport of elemental mercury, and a study has also confirmed that windborne transport may play an important role in the transport of pollutants [114]. Through previous studies, the enrichment of wild edible mushrooms for elemental mercury is influenced by a variety of factors, but mainly by the environment and by themselves [110,111,112,113]. Mushrooms grown near some mining and processing areas should be prevented from consumption. Mushrooms can also be used as a monitoring method for environmental contamination, and their substrates as components of the environment can demonstrate environmental contamination to some extent [113]. A comparative graph of mercury content is shown in Appendix A.

### 3.4. Physiological Role of Arsenic and Content of Arsenic in Wild Edible Mushrooms

Physiological role: Arsenic is classified as a metalloid, and its chemical properties are between metals and nonmetals. Arsenic is the most common element in environmental pollutants, and is mainly released into the environment by geological processes, metal smelting, and chemical manufacturing [115,116]. Arsenic is a carcinogen that comes into contact with humans through air, water, soil, and food. It can enter the body through respiratory inhalation, skin contact, and dietary intake, causing lung disease, reproductive problems, vascular disease, and gangrene [92,93,94]. After the absorbed arsenic enters the blood, 95–99% of the arsenic is combined with the globin of hemoglobin and then distributed to various organs, including the liver, lungs, kidneys, and skin [117]. The toxicity of inorganic arsenic will cause serious health hazards to both men and women, especially to reproductive function. Serious consequences for pregnant women may include miscarriage, stillbirth, or premature delivery. Pregnant women who are regularly exposed to arsenic in their lives or at work are more likely to miscarry or give birth to an underweight newborn because arsenic can enter the placenta and cause substantial damage. [90].

Zhang et al. [118] collected 48 species (Table 10) of edible mushrooms from Yunnan and Sichuan, China and the sampling sites were far from industrial contaminated areas. All samples were determined to have elemental arsenic values not exceeding 1.70 mg/kg. The authors compared the values of elemental arsenic in mushrooms from different countries and found that *Tricholoma* spp. differed greatly in their arsenic values in different environments [119,120]. The mushrooms collected from arsenic-contaminated areas had higher arsenic content. In contrast, the elemental arsenic content in mushrooms collected from uncontaminated areas was much lower [121]. Yildiz et al. [122] determined the elemental arsenic content in two cultivated and five wild edible mushroom species (Table 10) growing in Turkey. The elemental arsenic content in the cultivated mushrooms ranged from 0.018 mg/kg to 0.029 mg/kg, while in wild edible mushrooms it ranged from 0.2 mg/kg to 0.31 mg/kg. Wild edible mushrooms contain higher amounts of elemental arsenic than cultivated mushrooms, but none presents a health risk. The same conclusion was reached in another study, where Sun et al. [123] collected sclerotia of cultivated and wild *Wolfiporia extensa* from 41 uncontaminated rural areas of Yunnan, China. All samples had arsenic values ranging from 5.27 ± 4.33 ng/g to 161 ± 10.6 ng/g, and the authors conducted a health-risk assessment and concluded that there was no health risk to humans from consumption of the study samples. Sima et al. [124] collected four species (Table 10) of wild edible mushrooms in the Czech Republic and determined the elemental arsenic content. The results showed that only *X. chrysenteron* contained arsenic values up to 6.14 mg/kg, while the other three species were below 1 mg/kg. By comparing the bioconcentration factors, *C. rhacodes* had the strongest ability for arsenic element enrichment. The effect of substrate on elemental accumulation of different species was not significant and was mainly influenced by the species. A total of 37 species (Table 10) of wild edible mushrooms were collected by Vetter in 15 habitats in Hungary [119]. Among them, *L. amethysthea* had the highest value of 146.9 mg/kg of elemental arsenic, while the maximum value of *L. laccata* under the same genus was only 16.4 mg/kg. There is a great variation in the elemental arsenic content of different species in the same habitat, and there is a phenomenon that species under a certain genus always have higher concentrations of elemental arsenic than species of other genera. Some taxa can accumulate higher concentrations of arsenic without any significant relationship with habitat. For example, *Agaricus*, *Calvatia*, *Collybia*, *Laccaria*, *Langermannia*, *Lepista*, *Lycoperdon*, and *Macrolepiota* have better arsenic-enrichment ability. Studies also concluded that the high level of arsenic elements in wild edible mushrooms is mainly influenced by the environment and species of mushrooms [118,119,123,124]. When analyzing the health risk of mushrooms, various factors need to be considered, and the evaluation cannot be based on the content alone. For some edible mushrooms with high arsenic enrichment, consumers and relevant authorities should pay attention to them [119,124]. A comparative graph of arsenic content is shown in Appendix A.

## 4. Influencing Factors of Element Contents in Wild Edible Mushrooms

The content of the same mineral element is different in different species of edible mushrooms, whether cultivated or wild [122]. Many researchers have studied and analyzed this aspect in terms of human factors, processing methods, species, and so on, hoping to find influencing factors of mineral element content in edible mushrooms. Due to the rapid development of society, people have caused great harm to the environment while relying on the power of nature. Soil pollution, air pollution, water pollution, other great harm to animals and plants and the growth of edible mushrooms cannot be separated from the cultivation of nature, but in the polluted natural environment, it is difficult to avoid the invasion of heavy metal elements, and edible mushrooms are highly enriched in heavy metal elements. This fact makes people have great concerns about edible mushrooms, so it is of far-reaching significance to study the factors influencing the content of mineral elements in edible mushrooms. This section analyzes human factors, processing methods, strains, and so on to understand the factors that can affect the content of mineral elements in edible mushrooms.

### 4.1. Influence of Human Factors on Mineral Elements of Wild Edible Mushrooms

In the long history of human development, the gift of nature is the most primitive power of mankind. Human beings have today’s situation due to nature. Humans blindly seeking development has led to the environment being greatly damaged. Yangzhong is one of the fastest-developing regions in China. Since 1997, Yangzhong has changed rapidly from a traditional agricultural economy to an industrial economy. The process of transformation gave birth to several industries, such as metallurgy and chemicals, which produced all kinds of toxic substances that were released into nature [125]. All countries around the world cannot develop without these industries, and all the pollutants produced are left to nature for slow decomposition. This has led to the accumulation of harmful elements in varying degrees during the growth of edible mushrooms growing near metallurgical plants, chemical plants, industrial areas, etc. This phenomenon has led to consumers loving and hating edible mushrooms, as the safety risks involved cannot be ignored. As a result, some researchers are keen to determine the metal content of edible mushrooms growing near such areas. Mleczek et al. [126] determined the differences in elemental content of *Boletus badius* (note: this is a synonym of *Imleria badia)* grown in uncontaminated acidic sandy soil and contaminated alkaline flotation tailing sites in Poland. The results showed that the accumulation of most elements in the samples grown in the contaminated area was higher than in the samples grown in the uncontaminated area. Svoboda et al. [127] collected 56 samples of 23 wild-grown edible mushrooms from a heavily polluted area in eastern Slovakia from 1997 to 1998, and determined the element content of the fruit bodies by atomic absorption spectrometry. In the past, this area was seriously polluted by polymetallic ore mining and smelting and mercury smelters and copper smelters from 1963 to 1993. The results showed that, by comparing with the background value, the accumulation of mercury in fruit bodies of edible mushrooms was the most serious, followed by the accumulation of cadmium. Mercury is highly accumulated in the fruit bodies of *Suillus luteus, Hydnum repandum, Russula vesca, Ramaria aurea* and *Lycoperdon perlatum*. Cadmium is highly accumulated in *Xerocomus chrysenteron* and *Lycoperdon perlatum*, which are widely used in daily life. Lead and copper are highly accumulated in *Lycoperdon perlatum* and *Marasmius oreades*. Arvay et al. [113] collected samples of edible mushrooms in the central area of Spiš, and collected six kinds of edible mushroom samples (*Boletus pulverulentus, Cantharellus cibarius, Lactarius quietus, Macrolepiota procera, Russula xerampelina, Suillus grevillei*) from 2012 to 2013. The collection area is the most seriously affected area by emissions from mercury smelters, and the industrial development in this area has a long history, which can be traced back to the second half of the 19th century. During industrial development, millions of tons of mercury were discharged into the environment. The authors determined the mineral element content of edible mushrooms of six kinds, and found that the lowest mercury element content of these edible mushrooms fruit bodies was 0.09 mg/kg, and the highest even reached 471 mg/kg. Svoboda et al. [106] measured the mineral element contents of edible mushrooms growing in a historical Rudolfov silver-mining area to investigate the contents of mercury, cadmium, and lead in common edible mushrooms fruit bodies growing in forests near silver-mining areas in history. The background values of mercury, cadmium, and lead were obtained from polluted areas, and the contents of mercury, cadmium, and lead in fruit bodies of edible mushrooms measured were several times to dozens of times higher than the background values, so it can be seen that the metal-mining area had a certain influence on the content of mineral elements in edible mushrooms.

Environmental pollution caused by human activities has enriched a large number of heavy metals in the fruit bodies of edible mushrooms in the growth process [100,106,111,112,119,122]. Picking edible mushrooms should be avoided near mines, smelters, highways, and other places. The edible mushrooms collected in these areas are likely to be enriched in a large number of heavy metal elements, and there are certain risks if they are eaten [105,106,107,112].

### 4.2. Effect of Processing Methods on Mineral Element Content of Wild Edible Mushrooms

In an investigation of the processing methods of edible mushrooms, Ziarati et al. [128] conducted experiments on *A. bisporus* of Iran, compared and analyzed the essential mineral elements contents of raw *A. bisporus*, cooked *A. bisporus*, fried *A. bisporus*, and microwaved *A. bisporus* and found that the average contents of zinc, copper and iron in fried *A. bisporus* increased. The average contents of zinc, copper, and iron in *A. bisporus* treated by microwave significantly decreased. After the *A. bisporus* was fried again, the manganese content increased again, but after microwave treatment, the manganese content decreased. After further analysis, it was concluded that the content order of manganese and copper in fruit bodies of *A. bisporus* after different processing methods was fried > cooked > raw > microwave. The order of iron content was fried > raw > cooked > microwaved, zinc content fried > cooked > microwaved > raw, and magnesium and calcium raw > fried > cooked > microwaved. It can be seen that different cooking methods have a certain influence on the mineral element content of *A. bisporus*. The content range of Fe, Cu, Zn, and Mn in the fried state tends to be higher, all of which are higher than that of *A. bisporus* in the raw state. However, the magnesium and calcium contents of *A. bisporus* in the raw state are higher than those in other states. In another article published by Ziarati et al. [129], the mineral element content of fruit bodies of *A. bisporus* (conventional form and sliced form) was also measured and studied after different processing methods, which was different from the previous one in that the processing method was frozen. In 2013, the authors purchased 580 samples of *A. bisporus* from the six most famous packaging brands in the trustworthy market of Tehran, Iran for three consecutive seasons (winter, spring, and summer). The content of mineral elements in samples was determined under 10 different processing methods (raw and raw + freezing, cooked and cooked + freezing, frying and frying + freezing, microwaved and microwaved + freezing, slicing and slicing + freezing). The results showed that the processing method of frying increased the average contents of zinc, copper, and iron in all samples, but freezing after frying significantly reduced the contents of these elements (*p* < 0.01). The results also showed that the heating processing method increased the contents of copper, manganese, iron, zinc, cadmium, and lead in *A. bisporus* fruit bodies, and the average contents of the studied elements were reduced after different processing methods and freezing. In the analysis of the influence of different processing methods on the content of mineral elements in the fruit bodies of *Lentinus edodes*, Lee et al. [130] used steaming, ironing, microwave, boiling, and roasting to process the fruit bodies, and thus obtained the true retention value of mineral element content after different processing methods. The results showed that compared with other processing methods, water cooking would cause the most loss of mineral elements in fruit bodies, followed by hot scalding and steaming. Microwaving and baking can retain mineral elements in fruit bodies better than boiling, scalding, and steaming. The results show that the loss of mineral elements depends on the type of processing rather than the type of mineral elements. It is concluded that microwaving or baking can prevent the loss of mineral elements in *Lentinus edodes* during processing. In a study of cadmium in the fruit bodies of *Cantharellus cibarius* mushrooms, Drewnowska [131] found that pickling can greatly reduce the cadmium contamination of the fruit bodies. The content of cadmium in fresh fruit bodies of *C. cibarius* slices decreased 11 ± 7% and 36 ± 7% after blanching, while the content of cadmium in frozen fruit bodies decreased 40 ± 6%. After the blanched fresh fruit bodies were pickled, the cadmium content was further reduced by 42–71%. Blanching and pickling reduce 72–91% of the total cadmium content of the original content of the fruit bodies. It can be seen that the cadmium content in the fruit bodies of *C. cibarius* was significantly reduced by pickling.

From the above study, it is known that high-temperature processing (frying) can increase the average elemental content of mushrooms, while medium or low temperature is much less effective. This may be due to the fact that the elemental content of mushrooms is elevated due to the high water loss in the mushrooms. However, microwave treatment does not seem to follow this pattern, as in the study of Ziarati et al. [129], the elemental content of selenium, copper, and manganese after microwave treatment was lower than the elemental content of unprocessed mushrooms. The elemental contents of magnesium and calcium were lower than those of unprocessed mushrooms under frying, boiling, and microwaving, and again did not follow the pattern of increasing elemental content with water loss. It is concluded that the variation in elemental content values may be related to different processing methods. For reducing the elemental content of heavy metals, the most suitable way is high-temperature blanching.

### 4.3. Effect of Species on Mineral Element Content of Wild Edible Mushrooms

Vetter [132] collected fungal fruit bodies in Hungary to study the content of mineral elements. After excluding the possibility of obvious arsenic pollution or soil pollution, it was found that the arsenic concentration in most fungal groups was lower than 0.1 ppm. However, among *Tricholomataceae* spp., *Flammulina velutipes*, *Lepista nebularis*, and *Clitocybe inversa* had the lowest arsenic content of 5.38 ppm and the highest arsenic content of 14.69 ppm. In addition, the *Agaricus* genus, except *Agaricus silvaticus, A. abruptibulbus,* and *A. xanthoderma*, *A. augustus*, *A. purpurellus* and *Macrolepiota rhacodes* all contained a high concentration of arsenic, and the highest arsenic concentration of *Macrolepiota rhacodes* was 26.5 ppm. In 2014, Dimitrijevic et al. [133] measured the content of mineral elements in 11 species of the Boletaceae family from an unpolluted rural area (Jastrebac Mountain) near Nish town, Serbia, and the element concentrations were all determined based on the dry weight. There was no necessary connection between the content of the same element in different species of Boletaceae, and the element-enrichment capacity was not the same due to belonging to the same family. For example, the iron content of *Leccinum pseudoscaber* was 514.76 mg/kg, that of *Boletus rhodoxanthus* 74.02 mg/kg, and that of *Boletus fechtneri* much lower than those two kinds—only 24.40 mg/kg. This is only the difference in the content of iron. The harmful elements lead, cadmium, mercury, arsenic, etc. have also been determined by the authors. Different mushrooms have big or small differences. Therefore, in the same growth environment, the element content of edible mushroom fruit bodies will be affected by their own factors, and different strains lead to certain differences in element content. Elemental lead levels in 12 edible mushrooms from three uncontaminated areas in Spain were determined by Campos et al. [134]. The edible mushrooms collected from the three regions were the same (collected in October and November during an unusually wet autumn in 2006). The results showed that the content of lead was different, and *Cantharellus cibarius* contained the most lead. After research and analysis, the authors concluded that the ability of edible mushrooms to accumulate heavy metals (such as Nd, Pb, Th, and U) was not affected by the growing areas of edible mushrooms and their growing substrates. This is true in unpolluted areas, and the absorption of certain metals in edible mushrooms seems to be related to their species. Mendil et al. [63] collected a total of eight species of edible mushrooms in Kastamonu, Turkey and determined the contents of iron, manganese, copper, zinc, lead, and cadmium in their fruit bodies. The results showed that there were also different levels of the same element in different species of edible mushrooms. Mleczek et al. [135] determined the elemental content of edible mushrooms, trying to find factors affecting this. The experimental sample collection sites were almost over the whole Poland region. The research results of the authors showed that the accumulation of elements was different for different species of edible mushrooms. The same conclusion was also confirmed in Canakkale Province of Turkey, and Çayır et al. determined the element contents of three edible mushrooms fruit bodies: *Lactarius deliciosus, Russula delica,* and *Rhizopogon roseolus*. Among the three edible mushrooms, *R. delica* had the highest content of copper, zinc, lead, and cadmium, even if the sampling points were different. Taking lead as an example, the lowest and highest concentrations of Pb in *R. delica* collected at different sampling points were 0.59 and 3.05 mg/kg, respectively. It is suggested that this may be due to the difference in species, which leads to the difference in metal-accumulation ability of different species [136].

The distribution sites of edible mushrooms are wide, and there are some differences in the growth environment. This also leads to the fact that the mineral element content of edible mushrooms is somewhat affected and shows variable element enrichment. But the enrichment ability of edible mushrooms themselves for elements is also a reason affecting the mineral element content, and to confirm this notion correctly, many scholars have performed a large number of measurements of mineral elements of edible mushrooms growing in uncontaminated areas, and the results of related studies have also confirmed that the enrichment ability of the same element is different among different species in the same growth environment.

## 5. Conclusions

Summarizing previous studies, it was found that the differences of the four beneficial elements in wild edible mushrooms were mainly influenced by their own element enrichment capacity and environment. There are several conclusions, as follows: 1. the content of some elements in the soil substrate is positively correlated with the content of fruit bodies; 2. the ability of different species of elements differs—even if the concentration of some elements in the soil substrate is high, there is no high enrichment in the fruit bodies; 3. mushroom fruit bodies are affected by environmental pollution to some extent; 4. the elemental content in the mushroom caps may be higher than in the stipes. The reason may be that the caps were first exposed to atmospheric heavy metal deposition or heavy metals accumulated in the caps through bottom-up movement of the mycelium. The enrichment patterns of the four harmful heavy metals in edible mushrooms were approximately the same as those of the four beneficial elements. However, the enrichment of mercury in substrates was not necessarily influenced by the growth environment. This is because elemental mercury can be transported through the atmosphere and cause contamination. This leads to detectable mercury contamination in mushrooms grown in an environment free of elemental mercury contamination. Different processing methods can also have some effect on the elemental content of edible mushrooms. For example, blanching at high temperatures can better maintain the levels of iron, manganese, zinc, and copper and reduce the levels of cadmium. Although there are articles related to the study of the effects of elemental processing methods of wild mushrooms, the range of species studied is not large. The range of species could be expanded in the future to provide a reference for the edible mushroom processing industry and consumers’ daily consumption. In addition, pickers and consumers should be wary of wild edible mushrooms growing near industrial areas, highways, smelters, chemical plants, etc. These are not recommended for picking and consumption. Species with strong heavy metal enrichment capacity should be taken seriously and avoided as much as possible.

## Figures and Tables

**Figure 1 jof-08-00964-f001:**
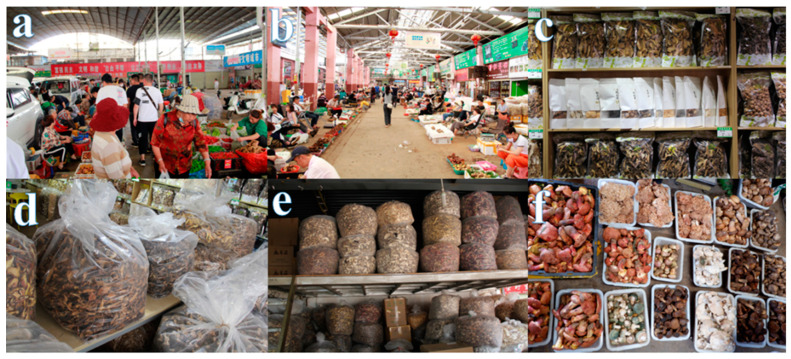
Scenes from a wild mushroom market in Mushuihua, Kunming, Yunnan Province, China (**a**,**b**). (**c**) Display of edible mushrooms in a dry retail shop. (**d**,**e**) Display of goods in a dry wholesale shop. (**f**) Display of some species of edible mushrooms.

**Figure 2 jof-08-00964-f002:**
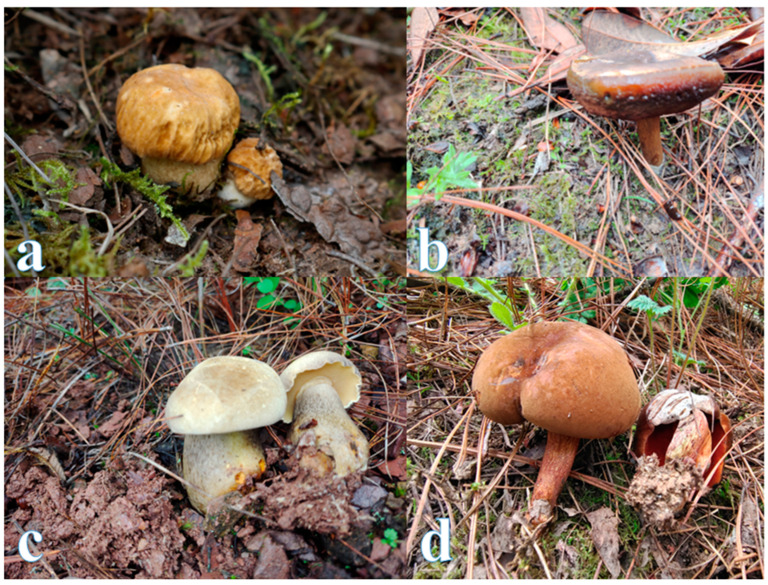
Wild edible mushroom growing environment. (**a**) *Boletus bainiugan*; (**b**) *Neoboletus brunneissimus*; (**c**) *Retiboletus fuscus*; (**d**) *Xerocomus* species.

**Figure 3 jof-08-00964-f003:**
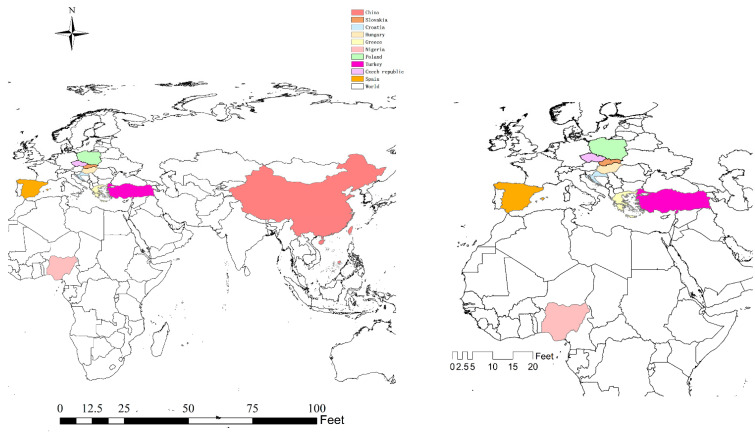
Geographical distribution map of countries covered.

**Table 1 jof-08-00964-t001:** Functional characteristics of essential trace elements for the human body.

	Main Existing Position	Symptoms	Function	References
Iron	Whole human body	Iron-deficiency anemia, liver development is blocked	An important component of human hemoglobin, myoglobin, and oxidase, which can solve symptoms such as iron deficiency anemia	[17,18,34]
Copper	Liver, muscles, and bone	Anemia symptoms, osteoporosis	Helps hemoglobin absorb iron and promotes the growth and formation of bone and myelin sheath	[16,35,36]
Zinc	Bone and skeletal muscle	Excessive zinc leads to nausea, vomiting, diarrhea, and fever, and deficiency of zinc affects growth, sexual maturity, and immune defense system	Essential component of a large number of zinc-dependent enzymes that facilitate the synthesis and degradation of lipids, proteins, and nucleic acids, among others	[37,38]
Manganese	Brain, kidney, pancreas, and liver	Deficiency leads to skeletal deformities, and excess causes human dysfunction	Protects the cardiovascular and cerebrovascular systems of the elderly and affects the body’s antiaging ability	[18,39,40,41]

**Table 2 jof-08-00964-t002:** Species, methods for determination of iron, and countries in the literature.

Species	Methods	Countries	References
*Agaricus bisporus, Polyporus squamosus, Pleurotus ostreatus, Armillaria mellea, Lepista nuda, Marasmius oreades, Boletus badius, Morchella esculenta, M. elata, M. vulgaris*	AAS (atomic absorption spectroscopy)	Turkey	[43]
*Cyathus striatus, Inonotus hispidus, Otidea onotica, Schizophyllum commune, Trichaptum biforme, Tricholoma fracticum, Xylaria polymorpha, Helvella elastica, Inocybe rimosa, Paxillus involutus, Amanita caesarea, A. ceciliae, A. vaginata, Agrocybe praecox, Cantharellus cibarius, Craterellus cornucopioides, Chroogomphus rutilus, Daedalea quercina, Fistulina hepatica, Gymnopus dryophilus, Ganoderma lucidum, Helvella crispa, Hydnum repandum, Infundibulicybe gibba, Macrolepiota procera, Tapinella atrotomentosa*	ICP-OES (inductively coupled plasma–optical emission spectrometry)	Turkey	[44]
*Cantharellus cibarius, Russula delica var chloroides, Ramaria largentii, Hygrophorus russula, Amanita caesarea, Fistulina hepatica, Boletus aureus, Armillaria tabescens, A. mellea, Lepista nuda*	AAS	Greece	[45]
*Agaricus bisporus, A. bitorquis, A gennadii, Coprinus comatus, Psathyrella candolleana, Volvopluteus gloiocephalus*	AAS	Greece	[46]

**Table 3 jof-08-00964-t003:** Species, methods for determination of copper, and countries in the literature.

Species	Methods	Countries	References
*Agaricus bisporus, Armillaria mellea, Boletus edulis, B. subtomentosus, Cantharellus cibarius, Cortinarius caperatus, Imleria badia, Lactarius deliciosus, Leccinum rufum, L. scabrum, Lentinula edodes, Macrolepiota procera, Pleurotus ostreatus, Russula heterophylla, R. vinosa, Suillus bovinus, S. grevillei, S. luteus, Tricholoma equestre, T. portentosum, Xerocomellus chrysenteron*	ICP-MS (inductively coupled plasma mass spectrometry)	Poland	[53]
*Morchella esculenta, Mitrophora semilibera, Agaricus bisporus, Lentinus strigosus, Coprinus comatus, Macrolepiota procera, Pleurotus eryngii, P. ostreatus, Lepista irina, L. nuda, L. personata, Agrocybe cylindracea, A. parecox, Pholiota aurivella, Volvariella bombycine, Leccinum scabrum, L. versipelle, Suillus luteus, Lactarius deliciosus, Russula delica*	AAS	Turkey	[54]
*Calvatia excipuliformis, Lycoperdon perlatum, Infundibulicybe gibba, Armillaria mellea, Marasmius oreades, Xerula radicata, Cantharellus cibarius, C. tubaeformis, Craterellus cibarius, Hypholoma fasciculare, Infundibulicybe gibba, Collybia dryophila, Lepista nuda, Mycena aetites*	GFAAS (graphite furnace atomic absorption spectrometry), AAS	Turkey	[55]
*Agaricus deliciosus, Boletus aereus, B. speciosus, Cantharellus cibarius, Catathelasma ventricosum, Dictyophora indusiata, Laccaria amethystea, Leccinum crocipodium, Lactarius crocatus, L. volemus, Polyporus ellisii, Russula aeruginea, R. alutacea, R. virescens, Ramaria botrytoides, Sarcodon imbricatum, Termitomyces albuminosus, Thelephora ganbajun, Tricholoma matsutake*	ICP-OES	China	[57]
*Agaricus campestris, Agrocybe aegerita, A. dura, Armillaria mellea, Boletus edulis, B. luteus, Coprinus comatus, Lactarius piperatus, L. salmonicolor, L. volemus, Marasmius oreades, Panellus stipticus, Piptoporus betulinus, Pleurotus ostreatus, Rhizopogon luteolus, Russula delica*	FAAS (flame atomic absorption spectrophotometry)	Turkey	[58]

**Table 4 jof-08-00964-t004:** Species, methods for determination of zinc, and countries in the literature.

Species	Methods	Countries	References
*Morchella vulgaris, M. esculenta, Helvella lacunosa, Agaricus campestris, A. urinascens, Coprinellus micaceus, Macrolepiota procera, Leucoagaricus nympharum, Boletus chrysenteron, Leccinum scabrum, Agrocybe dura, Cantharellus cibarius, Handkea utriformis, Lycoperdon perlatum, Vascellum pratense, Marasmius oreades, Pleurotus eryngii, P. ostreatus, Volvariella gloiocephala, Lentinus tigrinus, Polyporus squamosus, Psathyrella candolleana, Lactarius piperatus, Russula delica, Stropharia coronilla, Suillus granulatus, S. luteus, Infundibulicybe gibba, Lepista nuda, L. personata*	AAS	Turkey	[64]
*Cantharellus cibarius, Hydnum repandum, Lactarius salmonicolor, Xerocomus chrysenteron, Agaricus cupreobrunneus, Amanita franchetii, Hygrophorus chrysodon, H. eburneus*	AAS	Greece	[65]
*Agaricus campestris, Armillaria mellea, Boletus aestivalis, B. edulis, Clitocybe inversa, C. nebularis, Lactarius deterrimus, Macrolepiota procera, Tricholoma portentosum, T. terreum*	XRF (X-ray fluorescence spec trometry)	Croatia	[66]
*Lactarius salmonicolor*	AES (atomic emission spec troscopy)	Slovakia	[67]
*Agaricus bisporus, A. langei, Coprinus comatus, Hydnum repandum, Marasmius oreades, Armillaria ostoyae, Agrocybe praecox, Pleurotus eryngii, P. ostreatus, Cyclocybe cylindracea, Cantharellus cibarius, Clavulina cinerea, Leccinum scabrum, Suillus granulatus, S. luteus*	AAS	Turkey	[72]

**Table 5 jof-08-00964-t005:** Species, methods for determination of manganese, and countries in the literature.

Species	Methods	Countries	References
*Clitocybe alexandri, C. flaccida, Lepista inversa, Volvariella speciosa, Lactarius sanguifluus, L. semisanguifluus*	AAS	Turkey	[25]
*Morchella esculenta, Helvella leucopus, Pleurotus ostreatus, Lactarius deliciosus, Tricholoma terreum, Suillus luteus*	ICP-OES	Turkey	[73]
*Boletus edulis, B. luridiformis, Suillus granulatus, Amanita rubescens, Macrolepiota procera, Pleurotus ostreatus, Lepista nuda, Volvariella gloiocephala*	AAS	Greece	[74]
*Agaricus bisporus, A. bitorquis, A. essettei, Craterellus cornucopioides, Lepista nuda, Leucoagaricus leucothites, Ramaria flava*	ICP-MS	Turkey	[75]
*Tricholoma matsutake*	ICP-AES (inductively coupled plasma atomic emission spectrometry)	China	[76]

**Table 6 jof-08-00964-t006:** Functional characteristics of four heavy elements.

Elements	Main Concentrated Parts	Major Harms	Primary Contact Pathway	Excretion	References
Lead	Bones	Reproductive system inj- ury, preterm delivery, miscarriage, infertility in women, abnormal prostate function in men	Air, water, food	Urine, feces	[84,85,86]
Cadmium	Kidneys, liver, testes, etc.	The male reproductive function suffers damage, with some toxicity to the testes	Soil, food	Urine, feces	[19,87,88]
Mercury	CNS (central nervous system), digestive system, and kidneys, among others	Blood mercury levels higher than 8 mg/L reduce sperm quality and fertility, and cause damage to the respiratory system, skin, blood, and eyes	Food, air	Urine, feces	[89,90,91]
Arsenic	Liver, lung, kidneys, skin, etc.	Pulmonary disease, repro- ductive problems, vascular disease, gangrene. Preterm birth, miscarriage, stillbirth	Water, air, food, soil	Urine, feces	[90,92,93,94]

**Table 7 jof-08-00964-t007:** Species, methods for determination of lead and countries in the literature.

Species	Methods	Countries	References
*Agaricus campestris, A.macrosporus, Amanita rubescens, Boletus pinicola, B.badius, Cantharellus cibarius, Clitocybe nebularis, Coprinus comatus, Lactarius deliciosus, Lepista nuda, Macrolepiota procera, Russula cyanoxantha, Tricholoma portentosum*	GFAAS	Spain	[97]
*Boletus edulis, Xerocomus badius*	ICP-MS	Poland	[98]
*Chlorophyllum brunneum, Coprinus africanus, Pleurotus floridanus, Cantharellus cibarius, Cortinarius melliolens, Entoloma spp, Volvariella speciosa*	An Alpha-4 Cathodeon atomic absorption spectrophotometer	Nigeria	[100]
*Boletus aereus, Collybia albuminosa, Cantharellus cibarius, Agaricus blazei, Catathelasma ventricosum, Cordyceps militaris, Grifola frondosa, Thelephora ganbajun*	XRF	China	[101]

**Table 8 jof-08-00964-t008:** Species, methods for determination of cadmium, and countries in the literature.

Species	Methods	Countries	References
*Agrocybe aegerita, Agaricus bisporus, Coprinus comatus, Collybia velutipes, Clitocybe conglobata, Hypsizygus marmoreus, Hericium erinaceus, Lentinus edodes, Lepista sordida, Pleurotus nebrodensis, P.eryngii, P.ostreatus, Russula albida, Volvariella volvacea*	ICP-AES	China	[105]
*Cantharellus cibarius, Lepista nuda, Armillaria mellea, Collybia dryophila, Marasmius oreades, Amanita rubescens, Macrolepiota rhacodes, Agaricus silvaticus, Suillus bovinus, Leccinum scabrum, Xerocomus badius, X.chrysenteron, Lactarius piperatus, Russula cyanoxantha, Lycoperdon perlatum*	AAS	Czech Republic	[106]
*Pleurotus dryinus, Infundibulicybe gibba, Lycoperdon excipuliforme, L.perlatum, Paralepista flaccida, Psathyrella multipedata, L.perlatum, P. piluliformis (forest), Macrolepiota procera, I.gibba, P.piluliformis (grassland), Coprinus comatus, Hymenopellis radicata, Leucoagaricus leucothites, Agrocybe cylindracea, Lactarius deterrimus, Lepista nuda, Cratherellus cornucopioides*	HR-ICP-MS	Croatia	[108]

**Table 9 jof-08-00964-t009:** Species, methods for determination of mercury, and countries in the literature.

Species	Methods	Countries	References
*Boletus subtomentosus, Imleria badia, Xerocomellus chrysenteron*	CVAAS (cold vapor atomic ab sorption spectroscopy)	Slovakia	[110]
*Boletus reticulatus, Macrolepiota procera, Russula xerampelina, Suillus grevillei, Xerocomellus chrysenteron*	CVAAS	Slovakia	[111]
*Agaricus arvensis, A.esettei, A.campestris, A.haemorroidarius, A.silvaticus, Armillaria mellea, Clitocybe odora, Cratarellus cornucopioides, Fistulina hepatica, Hericium clathroides, Hirneola auricula, Hypholoma capnoides, Laccaria laccata, L.amethystina, Lactarius deliciosus, Lepista gilva, L.inversa, L.luscina, L.nebularis, L.nuda, Lycoperdon perlatum, Macrolepiota procera, M.rhacodes, M.mastoidea, Pleurotus ostreatus, P.pulmonarius, Suillus granulatus, S.grevillei, Stropharia aeruginosa, Tricholoma imbricatum, T.scalpturatum, T.terreum, Xerocomus chrysentheron, X.porosporus, X.subtomentosus*	AAS	Hungary	[112]
*Boletus pulverulentus, Cantharellus cibarius, Lactarius quietus, Macrolepiota procera, Russula xerampelina, Suillus grevillei*	FAAS/GFAAS	Slovakia	[113]

**Table 10 jof-08-00964-t010:** Species, methods for determination of arsenic and countries in the literature.

Species	Methods	Countries	References
*Albatrellus cristatus, Amanita caesarea, Amauroderma guangxiense, Auricularia auricula-judae, Boletus aereus, B.auripes, B.brunneissimus, B.edulis, B.ferrugineus, B.impolitus, B.magnificus, B.pallidus, B.satanas, B.speciosus, B.tomentipes, B.umbriniporus, Catathelasma ventricosum, Ganoderma capense, G.tsugae, G.philippii, Lactarius chichuensis, L.volemus, Leccinum rugosiceps, Lentinus edodes, Lepista nuda, Macrocybe gigantea, Osteina obducta, Pleurotus ostreatus, Ramaria formosa, R.rufescens, Retiboletus griseus, Russula pseudodelica, R.vinosa, R.violacea, R.virescens, Rigidoporus ulmarius, Sarcodon scabrosus, Scleroderma citrinum, Shiraia bambusicola, Sparassis crispa, Suillus pictus, Termitomyces globulus, Thelephora ganbajun, Tricholoma bakamatsutake, T.pessundatum, Wolfiporia extensa, Xerocomellus rubellus*	AFS (atomic fluorescence spec trometry)	China	[118]
*Amanita caesarea, Cantharellus cibarius, Craterellus cornucopioides, Fistulina hepatica, Meripilus giganteus*	ICP-MS	Turkey	[122]
*Wolfiporia extensa*	ICP-MC	China	[123]
*Chlorophyllum rhacodes, Suillus grevillei, Imleria badia, Xerocomellus chrysenteron*	—	Czech Republic	[124]
*Agaricus arvensis, A.augustus, A.campestris, A.esellei, A.langei, A.purpurellus, A.sylvaticus, A.sylvicola, Armillaria mellea, Calvatia excipuliformis, C.utriformis, Cantharellus cibarius, Clitocybe odora, Collybia butyracea, Craterellus cornucopioides, Fistulina hepatica, Hericium coralloides, Hirneola auricula, Hydnum repandum, Hypholoma capnoides, Laccaria amethysthea, L.laccata, Lactarius deliciosus, L.deterrimus, Langermannia gigantea, Lepista flaccida, L.nebularis, L.nuda, Lycoperdon perlatum, Macrolepiota procera, M.rhacodes, Pleurotus ostreatus, P.pulmonarius, Stropharia aeruginosa, Suillus granulatus, S.grevillei, Tricholoma terreum*	ICP-MS	Hungary	[119]

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
