# Peer review of "Research Progress on Elements of Wild Edible Mushrooms"

_jof, 2022, doi:10.3390/jof8090964_

Round 1

Reviewer 1 Report

Dear authors,

I received your manuscript under the title "Research Progress on Elements of Wild Edible Mushrooms" to review it. Based on my review, my proposal is to accept your paper after the minor changes that are listed in the pdf version of your manuscript that I am sending as an attachment.

Author Response

Dear reviewers, thank you very much for reviewing and judging my manuscript. Here I will answer your questions as follows:

Line 197--"You wrote here "brain", while in the second sentence you wrote that the highest Cu accounts is in the muscles and bones. So what is account of Cu in brain?"

Response: Thank you for your suggestion. “brain” has been modified into "muscle and bone". I have made the changes in 213 line, "Generally, the content of copper in normal adults is between 100 mg and 200 mg, mainly in the liver, blood, muscles and bones".

Lines 358-359--? please write clearly.

Response: Thank you for your suggestion. I have made the changes in lines 375-377, "Referring to previous studies, the elemental content of manganese is not considered high when compared to the elemental content of iron, copper and zinc ".

Lines 496, 508, 510, 597--The number of reference is missing.

Response: Thank you for your suggestion. I have corrected the reference positions.

Lines 718-719--What this means "raw/raw+freezing and etc? What / represents? Needs more explanation.

Response: Thank you for your suggestion. I have made the changes in lines 748-751, "The content of mineral elements in samples was determined under 10 different processing methods (raw and raw+freezing, cooked and cooked+freezing, frying and frying+freezing, microwave and microwave+freezing, slicing and slicing+freezing)".

Reviewer 2 Report

Thank you very much for your interesting research. Some points must be carefully revised:

Line 40. In many countries, vegetarians can access to animal proteins through eggs and dairy products. Perhaps it should be rephrased considering “vegans” instead of “vegetarians”.

Line 50-52. Other interesting functions such as hypocholesterolemic activity must be included in this paragraph. Useful reference: 10.1039/c8fo01704b. I consider that the most studied bioactive compounds (in general, not only with hypocholesterolemic activity, but those responsible of the cited activities) must be at least mentioned: beta-glucans, phenolic compounds, ergosterol, etc. Useful reference: 10.1615/IntJMedMushrooms.2017024413

Line 54. Typo: “musooms”

Lines 60-63. What about the top species in the mushrooms market? Could you include some related data?

Table 1. “All parts of the human body” might be replaced by “whole human body”

Line 151. Are the contents referred to dry weight?

Line 160. The genus must be written if it is the 1st appearance in the text.

Line 191-192. Is there a reason for this weaker understanding?

Line 452-453. Are they “naturally” heavy metals-rich? Or is it referred to enriched fractions/extracts or fortified mushrooms?

Line 695. You did a great specific revision about the effects of culinary treatments. Have you found some specific effects related to preservation treatments? E.g.: modified atmospheres or irradiation (electron beam, gamma, UV).

Author Response

Dear reviewers, thank you very much for reviewing and judging my manuscript. Here I will answer your questions as follows:

Line 40--In many countries, vegetarians can access to animal proteins through eggs and dairy products. Perhaps it should be rephrased considering “vegans” instead of “vegetarians”.

Response: Thank you for your suggestion. I have made the change in line 40.

Line 50-52--Other interesting functions such as hypocholesterolemic activity must be included in this paragraph. Useful reference: 10.1039/c8fo01704b. I consider that the most studied bioactive compounds (in general, not only with hypocholesterolemic activity, but those responsible of the cited activities) must be at least mentioned: beta-glucans, phenolic compounds, ergosterol, etc. Useful reference: 10.1615/IntJMedMushrooms.2017024413.

Response: Thank you for your suggestion. I have used both valuable references in my article and have made the changes in lines 52-55, “Mushrooms have been shown to have medicinal properties, and studies have shown that the beta-glucan in mushrooms may have a cholesterol-inhibiting effect and may have hypocholesterolemic activity [13, 14]”.

Line 54--Typo: “musooms”

Response: Thank you for your suggestion. I have made the change in line 58.

Lines 60-63--What about the top species in the mushrooms market? Could you include some related data?

Response: Thank you for your suggestion. I have made the changes in lines 64-68, “The main representative product of the wild edible mushroom market in Yunnan Province, China is the Boletaceae. According to the China Edible Mushroom Association, in 2020, the production of Boletaceae in Yunnan Province was 89,363.2 tons, with a total output value of RMB 281,261,82,000. (For more information check the website: http://bigdata.cefa.org.cn/output.html)”.

Table 1--“All parts of the human body” might be replaced by “whole human body”

Response: Thank you for your suggestion.  I have made the change in Table 1.

Line 151--Are the contents referred to dry weight?

Response: Thank you for your suggestion. I have made the changes in line 125, “Note: The elemental content values mentioned in the article are all dry matter content”.

Line 160--The genus must be written if it is the 1st appearance in the text.

Response: Thank you for your suggestion. Amanita is not the first appearance in line 160.

Line 191-192--Is there a reason for this weaker understanding?

Response: Thank you for your suggestion. I have made the changes in lines 204-208, “Compared to elements such as iron and zinc, no special attention has been paid to physical function problems caused by copper deficiency, resulting in relatively weak knowledge about copper”.

Line 452-453--Are they “naturally” heavy metals-rich? Or is it referred to enriched fractions/extracts or fortified mushrooms?

Response: Thank you for your suggestion. I have made the changes in lines 477-479, “We should avoid eating and picking mushrooms that are grown in environmentally contaminated areas and have a high capacity for heavy metal enrichment due to the characteristics of the species”.

Line 695--You did a great specific revision about the effects of culinary treatments. Have you found some specific effects related to preservation treatments? E.g.: modified atmospheres or irradiation (electron beam, gamma, UV).

Response: Thank you for your suggestion. The literature I refer to is mainly about the effect of processing on the elemental content of mushrooms, and does not address information on the preservation of mushrooms.

Reviewer 3 Report

attached

Author Response

Dear reviewers, thank you very much for reviewing and judging my manuscript. Here I will answer your questions as follows:

Lines 744-750--There are a number of paragraphs, which call for a more in-depth discussion. For example, on page 23 the authors vaguely concluded “that it is possible to realize the influence of some mineral element content of edible mushrooms through different processing treatments.” To me it became quite clear that processing methods, which are inevitably linked to a certain degree of shrinking (like frying) result in a decrease of even not so well water soluble ions, simply because the fruiting bodies lose mass (water). Simple warming, for example by microwaving, cannot result in the same effect, as only very little leakage of water occurs.

Response: Thank you for your suggestion. I have made the changes in lines 776-787, “From the above study, it is known that the high-temperature processing method (frying) can increase the average elemental content of mushrooms, while the medium or low temperature is much less effective. This may be due to the fact that the elemental content of mushrooms is elevated due to the high-water loss in the mushrooms. However, microwave treatment does not seem to follow this pattern, as in the study of Ziarati et al. [129] the elemental content of selenium, copper and manganese after microwave treatment was lower than the elemental content of unprocessed mushrooms. The elemental contents of magnesium and calcium were lower than those of unprocessed mushrooms under frying, boiling and microwave processing, and again did not follow the pattern of increasing elemental content with water loss. It is concluded that the variation of elemental content values may be related to different processing methods. For reducing the elemental content of heavy metals, the most suitable way is high-temperature blanching.”

I also recommend to indicate in an appropriate way that metal ions are meant throughout the text, unless otherwise stated (for example for elementary mercury).

Response: Thank you for your suggestion. This article focuses on the essential trace elements of iron, copper, zinc, manganese and the four heavy metals cadmium, lead, arsenic and mercury contained in wild edible mushrooms. The meanings of the elements indicated in your question are outlined respectively in the article. However, I am not sure if I understand your question correctly. If you have any other suggestions, please feel free to contact me.

Abstract: lines 16-17--What is meant by “relevant departments”?

Response: Thank you for your suggestion. I have made the changes in line 17, “Entry and exit inspection and quarantine departments”.

Abstract: line 19--omit “for the first time”

Response: Thank you for your suggestion. “for the first time” has been deleted.

Introduction: lines 34-38: Why was the Czech Republic chosen as an example? The same refers to many, especially poorer countries in the world.

Response: Thank you for your suggestion. The main reason for choosing the Czech Republic as an example in the paper is that its domestic citizens pick wild edible mushrooms as “a hobby of the whole people", and the data show that on average each household consumes 5.6 kg of mushrooms a year. It can be seen that mushrooms have a high degree of influence on the whole country and are more representative compared to other countries.

Lines 42-44--Because edible mushrooms have a high composition .. ” show a relatively high concentration of …..     Please amend this incomplete sentence.

Response: Thank you for your suggestion. I have made the changes in lines 42-44, “This is because of the high branched chain amino acids obtained in mushrooms, which are generally found only in animal proteins”.

Lines 49-53--the way to fight the virus …”  what about vaccination?

Response: Thank you for your suggestion. I have made the changes in lines 52-55, “Mushrooms have been shown to have medicinal properties, and studies have shown that the beta-glucan in mushrooms may have a cholesterol-inhibiting effect and may have hypocholesterolemic activity”.

Lines 58, 78: artificially cultivated

Response: Thank you for your suggestion. I have made the change in lines 62, 78.

Line 64--Trace elements have been needed”   They still are needed

Response: Thank you for your suggestion. I have made the change in line 73.

Figure 2--Legend is incomplete, what are a, b, c, d?

Response: Thank you for your suggestion. I have made the changes in Figure 2, “a: Boletus bainiugan; b: Neoboletus brunneissimus; c: Retiboletus fuscus; d: Xerocomus sp.”.

Figure 3-- Scale is apparently wrong & superfluous. Enlarge to make all countries well visible.

Response: Thank you for your suggestion. I have made the changes in Figure 3.

Table 1--Correct formatting

Response: Thank you for your suggestion. I have made the changes in Table 1.

Table 2 to 5 and 7 to 10--Abbreviation of the common, recurrently cited analytical methods could be used to create space for the enumeration of the species, which would very much facilitate legibility.

Response: Thank you for your suggestion. I have made the changes in Table 2 to 5 and 7 to 10.

Table 6--Correct page break page 13-14

Response: Thank you for your suggestion. I have made the changes in Table 6.

Conclusion: first para is a repetition, omit

Response: Thank you for your suggestion. The first paragraph of the conclusion has been deleted.

Reviewer 4 Report

Abstract, line 16. Replace "which is widely concerned by consumers" by "which is of great concern to consumers"

Line 42–44. The phrase starting with "Because ..." is a sentence fragment, not a proper sentence. Rewrite.

Line 48. Replace “Eating it” by “Eating them”

Line 61. “and its scale, the scene is quite grand” needs editing.

Line 64. Replace “have been needed” by “are needed”

Line 66–67. Replace by “Heavy elements are always harmful to human health [19–21].”

Line 102. Replace “is rich” by “are rich”. However, this sentence needs rewriting, as the wild-growing mushrooms have not become rich in proteins and vitamins “in recent years”, have they?

Line 105. The sentence containing the words “as mushrooms that rich in heavy metals” needs rewriting.

Line 236. It should read “are two small cities”

Line 381. “Because they and heavy metals have similar properties” is not a sentence and is just hanging there. It needs to be combined with the previous sentence or deleted

Lines 384–385. “are common in several the following” is meaningless as it stands. Rewrite.

Line 390 and in some other places. “human bodies” should be changed to “the human body” for consistency with previous usage in this paper. It is also more correct English.

Line 410. “testes” is better than “testis” for consistency with its use in this manuscript

Line 423. Insert “the” before “hymenophore”

Line 431–432. Is “X. badiusXerocomus badius? If so, X. needs to be replaced by Xerocomus. In fact, if this is the same species as Imleria badia, entered in Table 3, this should be mentioned here, as consistency in the use of names is important.

Lines 444–445. Delete “respectively”. Rewrite the next sentence as “It was found that the caps were more likely to have a higher elemental content than the stipes.”

Line 480. What does “However, the other heavy metals were not exceeded only by one mushroom” really mean? Rewrite.

Line 494. “Edulis” should read “edulis

Line 568, Table 9. “Hungry” should read “Hungary”.

Line 623, Table 10. “Hungry” should read “Hungary”.

Line 643. Replace “which leads to the environment has been greatly damaged” by “which has led to the environment being greatly damaged”

Lines 648–651. The two sentences starting with “All countries” need to be edited and combined into one sentence.

Lines 656–657. Boletus badius needs to be in italics. More importantly, is this species a synonym of Picipes badius or of Imleria badia, which appears often in the Tables of this manuscript?

Line 667. Should read Suillus.

Line 672. Spiš

Line 673. “grevillei”, not “qrevillei”

Line 691–692. Rewrite sentence as “Picking edible mushrooms should be avoided near mines, smelters, highways, and other places.”

Line 697. It should read “of Iran” rather than “of Iranian”

Line 699. Delete the full stop (.) and change “And” to “and”

Lines 725–726. Replace the full stop after edodes by a comma, as the next three lines (starting with Lee et al.) are part of the same sentence.

Line 728. Delete “of Lentinus edodes”

Line 743. C. cibarius needs to be in italics.

Lines 748–749. Replace “through the existing literature conclusions,” by “from the existing literature,”

Line 752. Reference number is needed after Vetter

Line 755. I presume “Le Pista Nebulis”?? should read “Lepista nebularis”!

Line 756. “Inversa” should read “inversa

Line 757. I think “in Agaricus genus” should read “the Agaricus genus”. It is not clear what “other edible mushrooms” is referring to.

Line 762. I question “Even though they all belong to Boletus,”. Is Boletaceae meant? In any case, why should different species in the same family have the same elemental content?

Line 769. “their”, not “thier”

Lines 770–771. “Twelve kinds …” This is a sentence fragment, not a sentence. It needs to be combined with the sentence that follows it.

Line 775. It should be “Cantharellus”, not “Cantharelus”.

Line 780. Delete the full stop (.) after “Turkey” and replace “And” by “and”

Line 784. Delete “out”

Line 786. Replace “author” by “authors”. There are several other places in the manuscript where “author” should be changed to “authors”, because the papers referred to have multiple authors, not just a single author.

Line 789. Replace “Andrhizopogon” by “Rhizopogon

There are a lot of misspellings of the names of fungal species in the Tables. Correct names, taken from Index Fungorum,  are given here.

Table 2 Polyporus squamosus, Morchella elata, Inocybe rimosa, Cantharellus cibarius, Russula delica and Armillaria tabescens

Table 3 Laccaria amethystina, Clitocybe gibba (this is the same species as Infundibulicybe gibba listed in Table 2). Lactarius crocipodium is probably a typographical error. Is L. crocatus or L. croceus meant? Termitomyces albuminosus, Armillaria mellea. The species list for China [52] needs to be put into alphabetical order.

Table 4 Clitocybe gibba [=Infundibulicybe gibba], Lactarius deterrimus

Table 5 Leucoagaricus leucothites

Table 7 Coprinus africanus, Pleurotus floridanus, Cordyceps militaris

Table 8 Volvariella volvacea, Hypsizygus marmoreus, Lentinus edodes. The species list for China needs to be put into alphabetical order.

Table 9 Boletus reticulatus, Imleria badia, Fistulina hepatica, Hericium clathroides. The two species lists for Slovakia need to be put into alphabetical order.

Table 10 Agaricus sylvaticus, Agaricus sylvicola (these are the preferred orthographic variants)

Author Response

Dear reviewers, thank you very much for reviewing and judging my manuscript. Here I will answer your questions as follows:

Abstract, line 16-Replace "which is widely concerned by consumers" by "which is of great concern to consumers"

Response: Thank you for your suggestion. I have made the changes in lines 16-17, “This is also one of the important factors affecting the import and export of edible mushrooms, which is of great concern to consumers and Entry and exit inpection and quarantine departments”.

Lines 42–44--The phrase starting with "Because ..." is a sentence fragment, not a proper sentence. Rewrite.

Response: Thank you for your suggestion. I have made the changes in lines 42-44, “This is because of the high branched chain amino acids obtained in mushrooms, which are generally found only in animal proteins”.

Line 48--Replace “Eating it” by “Eating them”

Response: Thank you for your suggestion. I have made the change in lines 51-52, “Eating them every day can improve the immune system of the human body, to achieve the effect of resisting diseases”.

Line 61--“and its scale, the scene is quite grand” needs editing.

Response: Thank you for your suggestion. “and its scale, the scene is quite grand” has been deleted.

Line 64--Replace “have been needed” by “are needed”

Response: Thank you for your suggestion. I have made the change in lines 73-74, “Trace elements still are needed by the human bady and play an extremely important role in the living system”.

Lines 66–67--Replace by “Heavy elements are always harmful to human health [19–21].”

Response: Thank you for your suggestion. I have made the changes in lines 76-77, “Heavy elements are always harmful to human health [19-21]”

Line 102--Replace “is rich” by “are rich”. However, this sentence needs rewriting, as the wild-growing mushrooms have not become rich in proteins and vitamins “in recent years”, have they?

Response: Thank you for your suggestion. I have made the changes in lines 114-117, “Wild-grown edible mushrooms are rich in protein and vitamins, and can supplement essential trace elements for the human body, which makes edible mushrooms become hot-selling products in the market and are popular with consumers.”

Line 105--The sentence containing the words “as mushrooms that rich in heavy metals” needs rewriting.

Response: Thank you for your suggestion. I have made the changes in line 117, “However, as mushrooms that they may be rich in heavy metals”.

Line 236--It should read “are two small cities”

Response: Thank you for your suggestion. I have made the changes in lines 252-253, “NevÅŸehir and NiÄŸde are two small cities in the tourist area of Cappadocia, Turkey”.

Line 381--“Because they and heavy metals have similar properties” is not a sentence and is just hanging there. It needs to be combined with the previous sentence or deleted

Response: Thank you for your suggestion. “Because they and heavy metals have similar properties” has been deleted.

Lines 384–385--“are common in several the following” is meaningless as it stands. Rewrite.

Response: Thank you for your suggestion. I have made the changes in lines 401-403, “Heavy metals, such as the elements cadmium, lead, mercury and arsenic, can pose a threat to human physiological functions and affect health when they are present in the body above acceptable levels”.

Line 390 and in some other places--“human bodies” should be changed to “the human body” for consistency with previous usage in this paper. It is also more correct English.

Response: Thank you for your suggestion. I have carefully checked the manuscript to exclude mistakes.

Line 410--“testes” is better than “testis” for consistency with its use in this manuscript

Response: Thank you for your suggestion. I have made the changes in lines 432-433, “Cadmium element can also cause damage to male reproductive function and has certain toxicity to the testes”

Line 423--Insert “the” before “hymenophore”

Response: Thank you for your suggestion. I have made the changes in lines 442, 445 and 447.

Line 431–432--Is “X. badius” Xerocomus badius? If so, X. needs to be replaced by Xerocomus. In fact, if this is the same species as Imleria badia, entered in Table 3, this should be mentioned here, as consistency in the use of names is important.

Response: Thank you for your suggestion. I have made the changes in lines 451-455, “Orywal et al. [89] determined the elemental cadmium content of the two most commonly consumed mushrooms (Table 7) in Poland and the value of cadmium was higher in B. edulis (1.983 mg/kg) than in Xerocomus badius (1.154 mg/kg) (Note: It is the same species as Imleria badia in Table 3 and it is a synonym of Imleria badia.)”.

Lines 444–445--Delete “respectively”. Rewrite the next sentence as “It was found that the caps were more likely to have a higher elemental content than the stipes.”

Response: Thank you for your suggestion. “respectively” has been deleted and I have made the changes in lines 468-469, “It was found that the caps were more likely to have a higher elemental content than the stipes”.

Line 480--What does “However, the other heavy metals were not exceeded only by one mushroom” really mean? Rewrite.

Response: Thank you for your suggestion. I have made the changes in line 506, “However, other mushrooms do not contain only one element in excess of the standard”.

Line 494--Edulis” should read “edulis

Response: Thank you for your suggestion. I have made the changes in lines 521-522, “Comparing the three mushrooms, B.edulis had the highest elemental lead content, followed by X. badius and X. chrysenteron”.

Line 568, Table 9. “Hungry” should read “Hungary”. Line 623, Table 10. “Hungry” should read “Hungary”.

Response: Thank you for your suggestion. I have made the changes in Table 9 and Table 10.

Line 643--Replace “which leads to the environment has been greatly damaged” by “which has led to the environment being greatly damaged”

Response: Thank you for your suggestion. I have made the changes in lines 669-671, “Due to humans blindly seek development, which has led to the environment being greatly damaged”.

Lines 648–651--The two sentences starting with “All countries” need to be edited and combined into one sentence.

Response: Thank you for your suggestion. I have made the changes in lines 675-677, “All countries around the world cannot develop without these industries, and all the pollutants produced are left to nature for slow decomposition.”

Lines 656–657--Boletus badius needs to be in italics. More importantly, is this species a synonym of Picipes badius or of Imleria badia, which appears often in the Tables of this manuscript?

Response: Thank you for your suggestion. I have made the changes in lines 685-687, “Mleczek et al. [126] determined the differences in elemental content of Boletus badius (Note: It is a synonym of Imleria badia.) grown in uncontaminated acidic sandy soil and contaminated alkaline flotation tailings sites in Poland”.

Line 667--Should read Suillus.

Response: Thank you for your suggestion. I have made the changes in line 697, “Mercury is highly accumulated in the fruit bodies of Suillus luteus,”.

Line 672--Spiš

Response: Thank you for your suggestion. I have made the changes in line 701, “Arvay et al. [110] collected samples of edible mushrooms in the central area of Spiš,”.

Line 673--“grevillei”, not “qrevillei”

Response: Thank you for your suggestion. I have made the changes in line 703, “Suillus grevillei”.

Line 691–692--Rewrite sentence as “Picking edible mushrooms should be avoided near mines, smelters, highways, and other places.”

Response: Thank you for your suggestion. I have made the changes in lines 722-723, “Picking edible mushrooms should be avoided near mines, smelters, highways, and other places”.

Line 697--It should read “of Iran” rather than “of Iranian”

Response: Thank you for your suggestion. I have made the changes in lines 727-728, “Ziarati et al. [128] conducted experiments on A. bisporus of Iran,”.

Line 699--Delete the full stop (.) and change “And” to “and”

Response: Thank you for your suggestion. I have made the changes in line 730, “and microwave treated A. bisporus and found that the average contents of zinc,”.

Lines 725–726--Replace the full stop after edodes by a comma, as the next three lines (starting with Lee et al.) are part of the same sentence.

Response: Thank you for your suggestion. I have made the changes in line 758, “Lentinus edodes,”.

Line 728--Delete “of Lentinus edodes”

Response: Thank you for your suggestion. “of Lentinus edodes” has been deleted.

Line 743--C. cibarius needs to be in italics.

Response: Thank you for your suggestion. I have made the changes in line 775.

Lines 748–749--Replace “through the existing literature conclusions,” by “from the existing literature,”

Response: Thank you for your suggestion. Based on the previous reviewer's comments, I have revised the entire paragraph as follows.

“From the above study, it is known that the high-temperature processing method (frying) can increase the average elemental content of mushrooms, while the medium or low temperature is much less effective. This may be due to the fact that the elemental content of mushrooms is elevated due to the high-water loss in the mushrooms. However, microwave treatment does not seem to follow this pattern, as in the study of Ziarati et al. [129] the elemental content of selenium, copper, and manganese after microwave treatment was lower than the elemental content of unprocessed mushrooms. The elemental contents of magnesium and calcium were lower than those of unprocessed mushrooms under frying, boiling, and microwave processing, and again did not follow the pattern of increasing elemental content with water loss. It is concluded that the variation of elemental content values may be related to different processing methods. For reducing the elemental content of heavy metals, the most suitable way is high-temperature blanching.”

Line 752--Reference number is needed after Vetter

Response: Thank you for your suggestion. I have made the changes in line 797, “Vetter [132] collected fungal fruit bodies in Hungary to study the content of mineral elements”.

Line 755--I presume “Le Pista Nebulis”?? should read “Lepista nebularis”!

Response: Thank you for your suggestion. I have made the changes in lines 800-801, “Lepista nebularis”.

Line 756--“Inversa” should read “inversa”

Response: Thank you for your suggestion. I have made the changes in line 801, “and Clitocybe inversa”.

Line 757--I think “in Agaricus genus” should read “the Agaricus genus”. It is not clear what “other edible mushrooms” is referring to.

Response: Thank you for your suggestion. I have made the changes in lines 802-805, “In addition, the Agaricus genus, except Agaricus silvaticus, A. abruptibulbus, and A. xanthoderma, A. augustus, A. purpurellus and Macrolepiota rhacodes all contain a high concentration of arsenic, and the highest arsenic concentration of Macrolepiota rhacodes is 26.5 ppm”.

Line 762--I question “Even though they all belong to Boletus,”. Is Boletaceae meant? In any case, why should different species in the same family have the same elemental content?

Response: Thank you for your suggestion. I have made the changes in lines 808-811, “There is no necessary connection between the content of the same element in different species of the Boletaceae, and the element enrichment capacity will not be the same due to belonging to the same family.”

Line 769--“their”, not “thier”

Response: Thank you for your suggestion. I have made the changes in line 818, “fruit bodies will be affected by their own factors”.

Lines 770–771--“Twelve kinds …” This is a sentence fragment, not a sentence. It needs to be combined with the sentence that follows it.

Response: Thank you for your suggestion. I have made the changes in lines 819-822, “Elemental lead levels in 12 edible mushrooms from three uncontaminated areas in Spain were determined by Campos et al [134]. The edible mushrooms collected from the three regions were the same (collected in October and November during an unusually wet autumn in 2006)”.

Line 775--It should be “Cantharellus”, not “Cantharelus”.

Response: Thank you for your suggestion. I have made the changes in line 826, “and Cantharellus cibarius”.

Line 780--Delete the full stop (.) after “Turkey” and replace “And” by “and”

Response: Thank you for your suggestion. I have made the changes in line 832, “Turkey and determined”.

Line 784--Delete “out”

Response: Thank you for your suggestion. “out” has been deleted.

Line 786--Replace “author” by “authors”. There are several other places in the manuscript where “author” should be changed to “authors”, because the papers referred to have multiple authors, not just a single author.

Response: Thank you for your suggestion. I have made the changes in lines 707, 746, 816, 827 and 838.

Line 789--Replace “Andrhizopogon” by “Rhizopogon”

Response: Thank you for your suggestion. I have made the changes in line 841, “and Rhizopogon”.

There are a lot of misspellings of the names of fungal species in the Tables. Correct names, taken from Index Fungorum,  are given here.

Table 2 Polyporus squamosus, Morchella elata, Inocybe rimosa, Cantharellus cibarius, Russula delica and Armillaria tabescens

Response: Thank you for your suggestion. I have made the changes in Table 2.

Table 3-- Laccaria amethystina, Clitocybe gibba (this is the same species as Infundibulicybe gibba listed in Table 2).

 Lactarius crocipodium is probably a typographical error. Is L.crocatus or L.croceus meant? Termitomyces albuminosus, Armillaria mellea. The species list for China [52] needs to be put into alphabetical order.

Response: Thank you for your suggestion. I have made the changes in Table 3.

Table 4-- Clitocybe gibba [=Infundibulicybe gibba], Lactarius deterrimus

Response: Thank you for your suggestion. I have made the changes in Table 4.

Table 5-- Leucoagaricus leucothites

Response: Thank you for your suggestion. I have made the changes in Table 5.

Table 7-- Coprinus africanusPleurotus floridanusCordyceps militaris

Response: Thank you for your suggestion. I have made the changes in Table 7.

Table 8-- Volvariella volvaceaHypsizygus marmoreusLentinus edodesThe species list for China needs to be put into alphabetical order.

Response: Thank you for your suggestion. I have made the changes in Table 8.

Table 9-- Boletus reticulatusImleria badiaFistulina hepaticaHericium clathroides. The two species lists for Slovakia need to be put into alphabetical order.

Response: Thank you for your suggestion. I have made the changes in Table 9.

Table 10-- Agaricus sylvaticusAgaricus sylvicola (these are the preferred orthographic variants)

Response: Thank you for your suggestion. I have made the changes in Table 10.